# Evaluating the impact of atmospheric forcing and air-sea coupling on near-coastal regional ocean prediction

Huw W. Lewis[1], John Siddorn[1], Juan Manuel Castillo Sanchez[1], Jon Petch[1], John M. Edwards[1], Tim Smyth[2]

[1]Met Office, Exeter, EX1 3PB, UK
[2]Plymouth Marine Laboratory, Plymouth, PL1 3DH, UK

*Correspondence to*: Huw W. Lewis (huw.lewis@metoffice.gov.uk)

**Abstract.**

Atmospheric forcing applied as ocean model boundary conditions can have a critical impact on the quality of ocean forecasts. This paper assesses the sensitivity of an eddy-resolving (1.5 km resolution) regional ocean model of the North-West European shelf (NWS) to choice of atmospheric forcing and atmosphere-ocean coupling. The analysis is focused on a month-long simulation experiment for July 2014 and evaluation of simulated sea surface temperature (SST) in a shallow near-coastal region to the south-west of the UK (Celtic Sea and western English Channel). Observations of the ocean and atmosphere are
used to evaluate model results, with a particular focus on the L4 ocean buoy from the Western Channel Observatory as a rare example of co-located data above and below the sea surface.

The impacts of differences in the atmospheric forcing are illustrated by comparing results from an ocean model run in forcing mode using operational global-scale numerical weather prediction (NWP) data with an ocean model run forced by a convective scale regional atmosphere model. The value of dynamically representing feedbacks between the atmosphere and ocean state
is assessed through use of these model components within a fully coupled ocean-wave-atmosphere system.

Simulated SST show considerable sensitivity to atmospheric forcing and to the impact of model coupling in near-coastal areas. A warm ocean bias relative to in-situ observations in the simulation forced by global-scale NWP (0.7 K in the model domain) is shown to be reduced (to 0.4 K) through use of the 1.5 km resolution regional atmosphere forcing. When simulated in coupled mode, this bias is further reduced (by 0.2 K).

Results demonstrate much greater variability of both surface heat budget terms and near-surface winds in the convective scale atmosphere model data, as might be expected. Assessment of the surface heat budget and wind forcing over the ocean is challenging due to a scarcity of observations. It can however be demonstrated that the wind speed over the ocean simulated by the convective scale atmosphere agreed with the limited number of observations less well than the global-scale NWP data. Further partially-coupled experiments are discussed to better understand why the degraded wind forcing does not detrimentally
impact on SST results.

**1 Introduction**

The exchanges of heat and momentum across the air-sea interface are fundamental components of the climate system (e.g. Yu et al., 2012), and can play a significant role in the evolution of natural hazards (e.g. Wada et al., 2018). In oceanography, accurate representation of the surface heat budget and near-surface winds and momentum fluxes are essential boundary conditions for ocean models given that they drive the ocean energy and dynamics from the surface (e.g. Lellouche et al., 2018). Despite this, routine evaluation of the quality of the surface forcing of operational ocean forecast systems receives relatively little focus. To a large extent, this reflects the challenge of observing these quantities over the ocean compared with on land, and thereby limited availability of measurements for evaluation (Drechsel et al., 2012; Banta et al., 2018). This may also be a result of operational ocean forecast systems running in a 'forced-mode' approach, whereby the surface forcing is provided from an external source of atmosphere model data. Typically the evaluation of atmosphere forecast quality is separated, potentially in science and organisational scope, from research and development of ocean forecast systems. Evaluation of wind forcing for operational wave models has been more prevalent, given the strong sensitivity of wave predictions to their accuracy (Cavaleri et al. 2009).The development of fully coupled atmosphere-ocean modelling prediction systems provide both motivation and tools with which to better understand the impact of the surface forcing on operational ocean forecasts (e.g. Pullen et al., 2017). This paper discusses an application of a regional coupled system for a North-West European shelf (NWS) domain at km-scale resolution to assess the impact of atmospheric forcing resolution and air-sea feedbacks on the quality of ocean predictions. The study focusses on a near-coastal region as they represent complex environments where providing accurate predictions can be more challenging through the strong influence of land-sea contrasts on both atmospheric forcing and ocean models (e.g. Holt et al., 2017; Cavaleri et al., 2018).

The role of atmospheric forcing and coupling has been previously addressed at coarser scales in the context of regional climate modelling. For example, Béranger et al. (2010) compared ocean simulations of the Mediterranean forced by atmospheric data provided at horizontal resolutions of about 100 km and 50 km. They found an important influence of the higher resolution wind forcing in particular in driving a more realistic ocean circulation. At increased resolution, Akhtar et al. (2018) showed improved wind speed and turbulent heat flux simulations using a 9 km spacing atmosphere model relative to 50 km more typical of global climate modelling, and both improved by coupling between ocean and atmosphere. It was noted that radiation fluxes were slightly better represented at the coarser resolution however, due to poorer representation of cloud cover in the 9 km resolution simulations.

An evaluation of the influence of surface fluxes on regional ocean simulations of the Mediterranean Sea was also assessed by Lebeaupin Brossier et al. (2011), who found that improving the temporal resolution of the atmospheric forcing, as well as the spatial resolution over some coastal areas, significantly changed the variability of mesoscale ocean processes. In regions where increased resolution enhanced near-surface winds, ocean convection was shown to be increased, although when applying higher frequency forcing the convection was dampened through changes to ocean stratification. Schaeffer et al. (2011) demonstrated improved representation of ocean eddies in the Gulf of Lion with a change from 9 km to 2.5 km resolution wind

forcing, but little impact of temporal resolution. Of relevance to the NWS, Bricheno et al. (2012) found a reduction on wind speed errors of more than 10% when moving from use of a 12 km to 4 km resolution atmosphere forcing for a wave-ocean coupled system of the Irish Sea.

A number of studies using a range of km-scale regional coupled systems more typical of the scale of current operational ocean
forecast systems have reported that simulated atmospheric fluxes can be improved through representing air-sea interactions (e.g. see Pullen et al., 2017 for a review). For example, Carniel et al. (2016) and Licer et al. (2016) assessed the impact of coupling on components of the surface heat budget for different coupled simulations of the Adriatic Sea, and showed that much improved turbulent heat fluxes resulted in improved predictions of sea surface temperature (SST) relative to forced mode ocean simulations. Similar sensitivity was demonstrated by Bruneau and Toumi (2016) for the Caspian Sea. Gronholz et al.
(2017) showed improved SST prediction for the North Sea through use of a higher resolution regional atmosphere forcing rather than a global-scale analysis, and further improvement through coupling between atmosphere and ocean. The influence of improved wind forcing through wave-atmosphere coupling was demonstrated by Wahl et al. (2017) for a similar domain.

The implications of the choice of atmospheric forcing and air-sea coupling on ocean forecasts for the NWS are assessed in this paper using the UKC3 regional coupled system. Lewis et al. (2018b) described the system in detail and provided an initial
domain-wide assessment of the UKC3 ocean performance for month-long simulations in four different seasons. This study focuses on near-coastal results for one of those periods in July 2014. The focus on the July 2014 results in this paper is motivated by Lewis et al. (2018b) having identified the impact of coupling on SST simulations to be greatest during summer. The focus here on assessing the near-coastal response in particular is also in contrast to the overview of results from atmosphere, ocean and wave components across the whole domain described by Lewis et al. (2018b) to summarise the overall
system performance. A further limitation of the initial discussion by Lewis et al. (2018b) arises from their comparison of coupled results with control simulations designed to be most analogous to the current approach adopted in operational systems. For the ocean model, differences between coupled results and the ocean-only control run forced by global-scale NWP may arise both from representing air-sea interactions and from the scale and characteristics of the atmospheric forcing differing between the two configurations. An additional uncoupled control simulation is therefore introduced in this study in which the
regional ocean model is forced by the higher resolution convective scale regional atmosphere model forcing, but without feedbacks between atmosphere and ocean. Further details on the application of UKC3 in the current study is described in Sect. 2. Simulated SST and the different atmosphere forcing are compared with available in-situ measurements in Sect. 3 and conclusions drawn in Sect. 4.

## 2 Methods

**2.1 Ocean model configurations and atmospheric forcing**

This study makes use of the AMM15 (Atlantic Margin Model, 1.5 km horizontal grid resolution) ocean model configuration, as described in detail by Graham et al. (2018), and in use for operational oceanography across the North-West European shelf

(NWS) within the Copernicus Marine Environment Monitoring Service (CMEMS; Tonani et al., 2019, *this issue*). AMM15 uses the NEMO ocean model code (vn3.6_STABLE, r6232; Madec et al., 2016). The model domain is illustrated in Fig. 1(a), which shows the relatively shallow North-West European shelf and shelf-break bounding to the North Atlantic to the west. The forced mode and coupled implementations evaluated in this paper were documented in detail by Lewis et al. (2018b).

A number of forced and coupled simulations spanning a month-long period between 30 June and 31 July 2014 have been conducted. To highlight ocean model performance in a near-coastal environment, the subsequent analysis focusses on evaluation relative to in-situ observations over the ocean within a section of the model domain encompassing the Celtic Sea and surrounding south-western approaches to the UK (Fig. 1(b)). The on-shelf part of this region has water depths of order 50 to 100 m and is seasonally stratified from late-April until September and well mixed through the rest of the year.

A summary of the four simulation experiments considered is given in Table 1. All ocean simulations were initialised from the same initial condition, taken from the 30-year free-running AMM15 simulation documented by Graham et al. (2018). As described by Lewis et al. (2018b), the same lateral boundary conditions using ocean model output from the coupled GloSea5 seasonal prediction system at 1/4° horizontal resolution (MacLachlan et al., 2015) were applied in all simulations. The same climatological freshwater discharge data were also applied to all simulations (Graham et al., 2018). All experiments are

conducted in forecast mode without data assimilation in any regional components.

Experiments FOR_GL and FOR_HI are forced mode ocean model simulations, in which externally generated atmospheric forcing are applied via file input. This is the approach most typically used in operational ocean forecast systems (e.g. Tonani et al., 2019, *this issue*). In forced mode, variables describing the surface heat and water budget and near-surface wind computed on an external atmosphere model grid are applied as a surface boundary condition in NEMO using the 'flux formulation'

methodology (Madec et al., 2016). The wind stress is computed in NEMO from the 10 m wind speed forcing, based on Smith and Banke (1975). The FOR_GL and FOR_HI runs contrast in the spatial scales and temporal resolution of atmospheric information applied. In FOR_GL forcing data originating from a global-scale operational weather forecast using the Met Office Unified Model (MetUM) are interpolated onto the 1.5 km resolution ocean grid. For the period considered in this paper, the global MetUM forecast system used the Global Atmosphere (GA) and Global Land (GL) version 6.1 science configurations,

documented in detail by Walters et al. (2017a). Across the NWS, global data from this system were available at a horizontal spatial resolution of about 17 km, with radiation variables applied at 3 hourly and wind components at hourly intervals through the simulation. The ocean surface boundary condition in the global MetUM is provided by the daily OSTIA (Operational Sea Surface Temperature and Sea Ice Analysis; Donlon et al., 2012). Surface currents are assumed zero and a constant global value for the Charnock parameter of 0.085 is used.

By contrast, FOR_HI is forced by variables interpolated from a regional atmosphere configuration of the MetUM, equivalent to that used for regional-scale operational weather prediction at the Met Office (RA1; Bush et al., 2018). The regional atmosphere configuration has a variable resolution grid (Tang et al., 2013), with a region of regularly spaced cells across the UK at 1.5 km horizontal spacing (Fig. 1(a)), and stretching out to 1.5 km x 4 km cells towards the domain boundaries. The regional atmosphere domain extent matches that of the regional ocean configuration (Lewis et al., 2018b). At this atmosphere

model resolution convection is explicitly resolved and local details such as the model coastlines and orography impact on the meteorology (e.g. Clark et al., 2016). All atmospheric data from this convective-scale km- resolution system were applied to the ocean at hourly frequency. For the month-long regional atmosphere simulation considered here, the surface boundary condition to the atmosphere model was also provided by interpolation from the daily OSTIA, and kept constant for each 24 h period. As in the global NWP system, ocean surface currents are assumed zero and a constant value for the Charnock parameter of 0.011 is now assumed. Details of the RA1 regional MetUM configuration, and how they relate to the global-scale NWP configuration, are provided by Bush et al. (2019). One of the key differences, related to the horizontal grid resolution is that atmospheric convection is explicitly represented in FOR_HI, whereas its simulation is parameterised in FOR_GL. The treatment of solar and terrestrial radiation also differ between RA1 and GA6.1 configurations. The RA1 configuration is most analogous to that used in GA7, which has an improved treatment of gaseous absorption compared to GA6 which typically result in reduced clear-sky outgoing long-wave radiation and increased downwards surface flux (Walters et al., 2017b). A final key difference between the global and regional MetUM configurations is that the parameterisation of clouds in FOR_GL uses the PC2 prognostic scheme (Wilson et al., 2008) and in FOR_HI uses the Smith (1990) diagnostic cloud scheme. One advantage of the prognostic approach is that clouds can be advected away from where they were created, but the diagnostic scheme is still considered to provide better forecasts in mid-latitude regional atmosphere configurations (Bush et al., 2018).

Coupled experiments CPL_AO and CPL_AOW use the AMM15 ocean model configuration as part of the UKC3 dynamically coupled system (Lewis et al., 2018b). The MetUM atmosphere model component is the same as used in atmosphere-only mode to provide FOR_HI forcing (i.e. 1.5 km variable resolution grid and RA1 science configuration), but now coupled directly to the ocean using the OASIS3-MCT (Craig et al., 2017) libraries with all information exchanged at hourly frequency. The CPL_AO simulation involves only atmosphere and ocean components being coupled – with heat budget terms, surface wind stress components and the surface pressure field passed from atmosphere to ocean components, and the simulated SST and currents passed from ocean to atmosphere. The 'fully coupled' CPL_AOW simulation also incorporates coupling between both atmosphere and ocean models to the WAVEWATCH III (Tolman et al., 2004) spectral wave model, defined on the same model grid as AMM15. Additional exchanged variables in CPL_AOW include the wind forcing from atmosphere to wave, the Charnock parameter from wave to atmosphere, water level and currents from ocean to wave, and significant wave height, Stokes drift components, and wave-modified surface drag from wave to ocean model components.

## 2.2 In-situ observations and the Western Channel Observatory

Atmosphere and ocean model simulations are compared to in-situ observations obtained from the operational network of surface automatic weather stations, ships and drifting or moored ocean buoys that are routinely exchanged in near real-time over the World Meteorological Organization Global Telecommunication System (GTS). A representative distribution of the location of these sites across the Celtic Sea sub-region is shown in Fig. 1(b). In this study, model data are compared with point observations by considering a mean of model output in the 5 x 5 neighbourhood of grid cells nearest to a given observation site. While this will smooth out some of the very fine resolution detail evident in AMM15 ocean simulations. However it is

considered a more representative approach than using only the nearest grid cell to reduce the 'double penalty' effects common with evaluating high resolution atmosphere or ocean model results for which a slight spatial or temporal displacement in the prediction of resolved small scale features relative to observations can lead to apparent relative errors at both observed and simulated locations, although the characteristics of such features may be well captured (e.g. Mass et al., 2002).

Around the southern UK coasts, most routine ocean observations are provided by the WaveNet monitoring network (Centre for Environment, Fisheries and Aquaculture Science; Cefas, http://wavenet.cefas.co.uk) and the Channel Coast Observatory (http://www.channelcoast.org). A number of these in Fig. 1(b) are sites where SST and near-surface wind observations are co-located. Figure 1(b) also highlights how the majority of ocean observing sites are located within only a few kilometres of the coast, and are therefore most representative of near-coastal conditions.

This study also uses atmosphere and ocean observations from a number of different sensors co-located at the L4 site of the Western Channel Coast Observatory (WCO; Smyth et al., 2010. See also https://www.westernchannelobservatory.org.uk). L4 is located at 50° 15′N, 4° 13′W, about 6 km away from the southern England coast, where the sea is about 50 m deep. A variety of long-term records of physical ocean, atmosphere and marine biogeochemical observations are recorded at L4 (Smyth et al., 2014). Of interest here are the in-situ surface and depth profile temperature measurements from a CTD, air temperature and

wind speed measurements, and total and diffuse solar radiation measurements within the 400 – 2700 nm wavelength range using a SPN1 Sunshine Pyranometer.

## 3 Results

### 3.1 Domain-wide sea surface temperature (SST)

Figure 2 summarises the mean difference between ocean model SST and in-situ buoy observations across the AMM15 domain

(e.g. see Fig. 1(b) of Lewis et al. (2018) for locations) during July 2014. Also shown is the equivalent comparison between daily OSTIA (Operational Sea Surface Temperature and Sea Ice Analysis; Donlon et al., 2012) and in-situ observations. Statistics of the mean difference (MD) and root mean square difference (RMSD) relative to all observations across the month are listed in Table 2. Figure 2 highlights that all ocean simulations had a common initial condition, which for this case was on average about 0.8 K warmer than observed. A summer time warm bias relative to OSTIA was noted by Graham et al. (2018).

This warm difference is maintained throughout the month for the FOR_GL simulation, with MD over the month of 0.73 K. This is consistent with the AMM15 run used to provide the initial conditions also being forced with a global-scale meteorology and being a well spun-up ocean state (Graham et al., 2018), so that the bias inherited from the initial condition is maintained. By contrast, the mean difference is substantially reduced comparing FOR_HI to observations (MD = 0.40 K), with FOR_GL and FOR_HI results diverging within the first few days of the simulation. This indicates SST prediction for the NWS is

sensitive to the choice of meteorological forcing.

Further reduction of the SST bias seen in Fig. 2 when considering coupling between the regional ocean and atmosphere models in CPL_AO (MD = 0.26 K). There is some additional value evident from coupling information of the wave state to ocean and

atmosphere components in CPL_AOW (MD = 0.20 K), although this is of secondary importance to the impact of either changing the source of atmosphere forcing or ocean-atmosphere coupling for this period and location.

## 3.2 SST in the Celtic Sea

To further examine the sensitivity highlighted in Fig. 2, the remaining analysis focuses on results across the Celtic Sea region only, and considers simulation results over the 10-day period between 20 July and 30 July 2014, as being representative of the different ocean simulations having spun up sufficiently from the same initial condition. This is supported by the summary statistics considering only this region and period listed in Table 2, from which broadly consistent conclusions can be drawn as from the statistics obtained for the full domain and simulation duration. In this case, the MD for CPL_AOW is 1 K smaller than that for FOR_GL results, and the RMSD is reduced from 1.6 K to 1.0 K.

A snapshot comparison of SST across the Celtic Sea on 28 July 2014 from FOR_GL and FOR_HI simulations with OSTIA show qualitatively very consistent patterns (Fig. 3(a) and 3(b)). These snapshots are representative of the 10-day mean differences shown in Fig. 3(d) and 3(e). Areas of relatively cooler water are simulated around west-facing peninsulas such as the Ushant front region to the west of Brittany, and around south-western England. Simulated SST across much of the Celtic Sea is relatively cooler in FOR_HI than FOR_GL however, in closer agreement with OSTIA overall. Both simulations have warmer surface water in near-coastal regions than observed, such as in the Bristol Channel where the simulated SST exceeds 294.5 K on 28 July.

Instantaneous and 10-day mean SST from the coupled CPL_AOW simulation are shown in Fig. 3(c) and 3(f) respectively. There is an extensive region where SST is reduced by more than 0.5 K across the Celtic Sea. While differences are lower through the English Channel, stronger relative cooling is also apparent along the coastlines of southern Wales, within the Bristol Channel, and around the Isle of Wight to the east of the domain section. In general, the CPL_AOW results are in closer agreement with OSTIA (Fig. 2), although there is some compensation between the coupled model being relatively cooler in more open ocean and warmer in near coastal areas. Figure 3(g)-(i) compares the RMSD over 10 days for each simulation with in situ observations relative to the RMSD between OSTIA and observations at each site. This highlights the relatively poor agreement of FOR_GL results (Fig. 3(g)) but relative improvements in RMSD for CPL_AOW results by in excess of 20 % at all near-coastal observing sites (Fig. 3(i)).

SST results at L4 between 20 and 30 July 2014 are shown in Fig. 4(a). At this location, the coupled experiments are cooler than observed, although the lowest RMSD (of 0.5 K) is obtained for CPL_AOW. The SST observations at L4 during late July 2014 were highly variable, with an observed range of 4 K shown in Fig. 4(a). On several days (e.g. 20, 21, 23, 26 and 29 July) a tidally-dominated heating signal of about 1 K is apparent. This was particularly strong on 22 and 25 July, potentially linked to strong solar heating in additional to tidal influence, when a range of 2 K and 3 K were observed respectively. More synoptic-scale influences appear to dominate on 27 and 28 July when the observed SST cycle was relatively diminished. The temporal variability of SST at L4 for FOR_GL is in general larger on diurnal timescales than observed, but reasonably well captured by all other ocean simulations with high-resolution atmosphere forcing (Fig. 4(a)). This is not the case on 25 July however,

when the increase in FOR_GL temperature through the day matches the observed range, while all other simulations fail to replicate such strong temperature variation.

In addition to surface measurements, depth resolved temperature data are routinely taken using CTD sensors at the L4 site on days when data are manually collected. One such profile was observed during the morning of 28 July 2014, and is compared with daily mean simulated temperature profiles at L4 in Fig. 4(b). The observed profile shows a strong temperature gradient between depths of 10 m and 15 m marking the mixed layer depth (MLD), with well mixed water near the surface and stratified water below to the sea bed. There are substantial differences between the simulated profiles in Fig. 4(b). The excessive surface heating in FOR_GL can be attributed to a much shallower MLD than observed, such that any input solar heating at the surface will heat a smaller volume of water than in reality. In contrast, the near-surface temperature and MLD is in good agreement with observations on this day in the FOR_HI simulation with high resolution atmospheric forcing. The strength of cooling across the thermocline is considerably less sharp than observed (or in FOR_GL), although this may be partly an artefact of using a daily mean rather than instantaneous profile and of averaging simulation results across a 5 x 5 neighbourhood of grid cells. Mean temperatures from FOR_HI are order 1 K warmer than observed between the MLD and a depth of about 35 m. This mean difference is improved when the ocean and atmosphere are coupled (CPL_AO), reflecting a positive impact of representing air-sea interactions within the system both at (Fig. 4(a)) and below the surface (Fig. 4(b)). An improved temperature profile at L4 below the mixed layer in the fully coupled CPL_AOW simulation is offset by a cool surface bias, leading to a relatively weaker temperature transition than in CPL_AO. Further tuning of the CPL_AOW system may be appropriate, as discussed by Lewis et al. (2018c, *this issue*).

These results demonstrate that SST and temperature profiles through depth are particularly sensitive to the source of atmospheric forcing and to representation of air-sea interactions across the NWS, with fundamental differences in the vertical structure developing between simulations from a common initial condition over a relatively short period of time.

## 3.3 Surface heat budget

The ocean surface boundary condition characterising the heat budget in NEMO is expressed in terms of the solar radiation, $Q_{SW}$, that penetrates the top few metres of the ocean, and a non-penetrative component, $Q_{ns}$, which only heats or cools the surface (Madec et al., 2016). In the AMM15 configuration, $Q_{SW}$ specifies the net shortwave radiation at the surface simulated by an atmosphere model across all wavelengths, and $Q_{ns}$ is computed from the surface heat budget variables as,

$$Q_{ns} = Q_{LW} - \lambda E - H, \tag{1}$$

with $Q_{LW}$ denoting the net surface longwave radiation, $\lambda E$ the latent heat due to evaporation and $H$ the sensible heat flux. In NEMO, the fraction of $Q_{SW}$ which penetrates to lower depths is controlled by the *rn_abs* parameter. In the simulations considered in this study, it is assumed that 66% of radiation is absorbed at the surface (Lewis et al., 2018b).

The spatial distribution of $Q_{SW}$, $Q_{LW}$, $\lambda E$ and $H$ used as forcing for FOR_GL (i.e. interpolated from the global-scale operational MetUM) is shown as 10-day means in Fig. 5, together with the mean difference between FOR_HI (i.e. interpolated from the variable resolution regional atmosphere simulation) and FOR_GL. The magnitude of mean net solar short-wave

radiation of order 250 Wm$^{-2}$ (Fig. 5(a)) clearly dominates the heat budget relative to the net long-wave radiation (of order 50 Wm$^{-2}$ away from the surface, Fig. 5(b)) and sensible heat flux (mean 5 Wm$^{-2}$ away from the surface across the Celtic Sea, Fig. 5(c)). The latent heating over the ocean is also shown to be a relatively important contribution to the surface energy balance, with a mean of order 50 Wm$^{-2}$ in FOR_GL forcing (positive values indicating a flux of heat to the atmosphere from evaporation of sea surface water). Comparing the spatial distribution of FOR_HI and FOR_GL heat budget terms in Fig. 5(e)-(h) shows generally close agreement on the large-scale (noting the scale of differences relative to the flux magnitudes), particularly for the sensible and latent heating which are driven by near-surface variability, although the magnitude of latent heating in FOR_HI is larger than FOR_GL. A key difference is the reduced mean solar radiation $Q_{SW}$ in FOR_HI relative to FOR_GL by more than 25 Wm$^{-2}$ across the Celtic Sea (Fig. 5(e), and reduced long-wave radiation loss away from the surface (Fig. 5(f)). The local scale variability of heating is also substantially greater in FOR_HI than FOR_GL, as might be expected given the contrast in atmosphere model resolutions and representation of convection. An imprint of a pattern of convective cells can be seen in the FOR_HI forcing differences for example, which likely leads to highly variable heating in time.

The spatial distribution of time mean differences between CPL_AOW and FOR_HI heat budget terms between 20 and 30 July 2014 are shown in Fig. 6. The impact of coupling on $Q_{SW}$ and $Q_{LW}$ is dominated by random changes in the spatial distribution of convection (Fig. 6(a),(b)). Examination of the simulated cloud fields during this period (not shown) indicate substantial changes in the exact spatial distribution of clouds at any given time between FOR_HI and CPL_AOW for example. The clearest relative impact of air-sea coupling is on the latent heat flux, which is broadly reduced by order 20% across the Celtic Sea in CPL_AOW. There is also some evidence that the latent heat flux is increased in near-coastal regions in CPL_AOW relative to FOR_HI. This coincides with regions of cooler SST in CPL_AOW than FOR_HI (Fig. 3), and in closer agreement with in-situ observations.

The sunshine pyranometer sensor at L4 provides a rare source of observations of the solar radiation over the ocean (Fig. 7(a)). The raw measurements at 1 min sampling frequency have not been corrected for wave motion, which can lead to considerable variability, particularly when the sea state increases. The data shown in Fig. 7(a) are hourly mean values and therefore considered as being representative. The total observed solar radiation exceeds 800 Wm$^{-2}$ on several days between 20 and 30 July 2014, particularly on 20 – 23 July, but increased cloud cover on 24 July leads to the most of the observed radiation coming from the diffuse component at L4. Given that the observations cover the wavelength range 400 – 2700 nm, these are not directly compared with the atmosphere model data. The time series of simulated $Q_{SW}$ across all wavelengths at the L4 location (Fig. 7(b)) shows broad agreement however. On most days, the simulated peak in short-wave flux at L4 differ between the sources of atmosphere data considered within 100 Wm$^{-2}$ and with FOR_GL typically lower than the regional atmosphere data. The different temporal resolution of the data, with the 3 h updates of FOR_GL being insufficient to adequately capture the daytime maximum, is a possible explanation for the difference. The warm surface temperature bias of FOR_GL at L4 is therefore not readily explained by assessing the local radiation budget in the immediate vicinity . The global- and regional-scale data differ more on 24 July, when FOR_GL has much lower $Q_{SW}$, in good qualitative agreement with the L4 observations (Fig. 7(a)). The FOR_HI and coupled simulations all have a strong diurnal variation in contrast on this day. Despite this, the

rate of simulated SST change at L4 in Fig. 4(a) on this day was generally consistent across each simulation, suggesting this to be mostly tidally-driven rather than a result of local heating. Time series of the non-penetrating heat budget term $Q_{ns}$ are shown at L4 is shown in Fig. 7(c). Values typically agree within 50 Wm$^{-2}$ between experiments through the period, although it is interesting to note that FOR_HI data are more variable than either the global-scale FOR_GL forcing or the coupled system results.

Although it is particularly challenging to routinely measure all components of the surface heat budget over the ocean (Yu et al., 2012), the availability of both air and surface temperature observations at L4 enables at least some comparison of the near-surface stability profile (air – surface temperature) against the high-resolution atmosphere simulations (Fig. 7(d)). The magnitude of the observed diurnal variability is in general well captured by all simulations, although air-sea coupling appears to correct periods on 22, 23 and 29 July when the FOR_HI regional atmosphere simulation has surface temperature too warm relative to air temperature, which cause spikes in sensible heat flux that are reflected in the $Q_{ns}$ comparisons (Fig. 7(c)).

Taking a broader perspective of the surface heat budget across all sea areas in the Celtic Sea sub-region shows the net effect of the different atmosphere forcing and air-sea coupling (Fig. 8). In Fig. 8(a)-(c), variables are accumulated across all model grid cells over sea in the region and time series of the spatial standard deviations shown in Fig. 8(d)-(f). In contrast to Fig. 7(b) for the L4 site, the accumulated net radiation (sum of short-wave and long-wave) across the whole region in Fig. 8(a) shows more consistently increased net radiation in the FOR_GL data. On 22 July 2014 for example, the mean daytime maximum net radiation (not shown) is over 150 W m$^{-2}$ higher in FOR_GL than the high resolution data. Values are also consistently higher during night time in the global-scale forcing data. These differences are reflected in a mean net radiation flux over the 10 days shown of 244 W m$^{-2}$ in FOR_GL compared with 227 W m$^{-2}$ in the CPL_AOW simulation. The mean net radiation for the Celtic Sea is order 7% higher in FOR_GL data than any of the regional-scale runs. This difference is consistent with the warm SST bias of FOR_GL relative to FOR_HI or coupled ocean simulations being driven by a relatively higher net radiation when using the global-scale atmospheric forcing relative to the regional scale. Figure 5 illustrates the FOR_GL simulated heat budget terms to be relatively smooth fields while high variability of radiation between convective cells in FOR_HI and coupled simulations is thought to produce small scale areas of relatively reduced heating which contribute to the reduced short-wave radiation flux shown in Fig. 5(e) for example. Some evidence of this is apparent in the time series of net short-wave radiation at L4 on 28 July 2014 in Fig. 7(b). The effect of different atmosphere forcing and coupling  is also highlighted through considering the standard deviation of net surface radiation across the region (Fig. 8(d)). A summary of these results is given in Table 3, which shows that daytime maximum values in excess of 250 W m$^{-2}$ are calculated using either FOR_HI or coupled results. In contrast, the standard deviation of the FOR_GL radiation data are consistently lower during both day and night and with a maximum standard deviation of less than 200 W m$^{-2}$, but typically of order 20-50% lower than high resolution atmosphere simulation values (Fig. 8(d)).

The accumulated non-penetrating radiation term, $Q_{ns}$, (Eq. (1)) across the Celtic Sea (Fig. 8(b)) shows much smaller net differences between experiments than for $Q_{SW}$. Time series of the spatial standard deviation of  $Q_{ns}$ across the region in Fig. 8(e) also demonstrate greater variability for the regional-scale forcing, and larger  differences between FOR_HI and the

coupled simulations (with CPL_AO and CPL_AOW being more consistent with each other). The difference between global and regional-scale time series on 27 – 29 July can be attributed to the sensitivity to latent heat flux (Fig. 8(c)). The reduced latent heating due to coupling during this period also results in a less strong upward (i.e. less negative) $Q_{ns}$ for coupled results relative to FOR_HI in Fig. 8(b). Lebaupin Brossier et al. (2015) assessed the role of atmosphere-ocean coupling on the water budget of the Mediterranean simulated using a 20 km resolution regional atmosphere and 1/12° ocean model components, with SST found to be a key controlling factor of evaporation. This link can also be clearly seen in the Celtic Sea by the clear spatial similarity between the impact of coupling on latent heating in Fig. 6(d) with the difference between the mean CPL_AOW SST field and OSTIA in Fig. 3(f) – noting that OSTIA data were used as the SST boundary condition driving the FOR_HI atmosphere simulations. In summary, the key sensitivity of the regional ocean simulations to differences in the surface heat budget from different sources of atmospheric forcing is dominated by the representation of the net short-wave radiation. A second-order but non-negligible differences in the latent heat flux is also found, linked to the different representation of SST in atmosphere simulations. When using a global-scale atmosphere forcing, as typical for most operational ocean forecast systems, the high spatial variability associated with convection is not captured, which leads to a larger accumulated heating over a given region in this case. Applying a more spatially variable representation of the surface heat budget when using the regional-scale forcing (FOR_HI) or atmosphere-ocean coupled systems (CPL_AO or CPL_AOW) contributed to the improvement to the warm SST bias found in the FOR_GL ocean simulation.

### 3.4 Near-surface wind speed

Snapshots of the global-scale and high-resolution regional atmosphere model wind speed at 10 m above the surface in Fig. 9 also reflect the much finer convective structures simulated in the FOR_HI simulations (Fig 9(b)). The general structure of wind speed available from the operational global-scale MetUM atmosphere model (Fig. 9(a)) is in qualitative agreement with in-situ observations at this time, particularly in reflecting areas of reduced wind speed across the Bristol Channel and off the southern England coast. The observations over sea are spatially more variable than FOR_GL across the region however. In contrast, the FOR_HI data show an area of strong convective activity over the Celtic Sea, and the spatial variability of wind speed over the ocean qualitatively appears to be as high as over land (Fig. 9(b)). The impact of coupling, quantified as the mean difference over the 10-day period between 20 and 30 July 2014 (Fig. 9(f)), shows wind speed differences of ±0.5 ms$^{-1}$, largely focused in the English Channel rather than Celtic Sea.

The atmospheric forcing and coupled results are compared with near-surface wind speed observations at L4 in Fig. 10(a). This shows results typical of that found at other sites in the region (Fig 10(b)) and more generally from analysis of a number of case studies by Lewis et al. (2018a, 2018b) for example. FOR_GL data follow the day-to-day variability of observed wind speed closely (MD = -0.07 ms$^{-1}$, RMSD = 1.29 ms$^{-1}$). By contrast, all high resolution experiments are biased fast (e.g. MD = 1.4 ms$^{-1}$ for CPL_AOW) and with increased RMSD relative to observations (Fig. 10(b)). The high temporal variability of wind speed also appears to exceed the observed variability. Figure 11 summarises the mean and range of differences between the global-scale forcing and CPL_AOW simulations relative to all observations across the Celtic Sea region. The wind speed bias in

CPL_AOW (and other regional atmosphere data, not shown) becomes particularly high on 27 July. The summary metrics indicate that both CPL_AO and CPL_AOW simulations have reduced differences to observations than FOR_HI during the period, although the influence of wave coupling feedbacks is generally small at this time of year. Figure 11(c) and Table 3 summarise the enhanced wind speed variability with increased model resolution in terms of the standard deviation of values across the Celtic Sea region for the regional scale data relative to FOR_GL.

Given the strong sensitivity of surface waves to the near-surface winds, the different characteristics of simulated winds between global and regional-scale systems has been found to have a detrimental impact of the quality of wave model simulations when forced with high-resolution data (Lewis et al., 2018a). As demonstrated in Fig. 10(a), this can be mitigated to some extent through coupling, but it remains challenging to improve the quality of wave forecasts relative to a system with global-scale forcing.

### 3.5 Partially coupled sensitivity experiments

Further work is clearly required to better understand and improve the quality of near-surface winds in the regional atmosphere model. It is therefore of interest to note that the quality of SST from the FOR_HI and coupled ocean simulations was improved relative to FOR_GL, perhaps despite the change in wind speed characteristics.

Two additional ocean-atmosphere coupled simulation experiments have therefore been conducted to further assess the impact of the heat budget and wind speed forcing changes on the ocean simulation. In pCPL_WIN, only the wind speed components are coupled between the atmosphere and ocean, and radiation variables read from the operational global forcing. In pCPL_RAD, only the radiation variables are coupled and the global-scale wind speed forcing is used. In both simulations, the exchange of variables and feedback from the ocean to the atmosphere was the same as in CPL_AO. Note that these partially coupled simulations are conducted to help attribute the relative impact of energy balance and near-surface wind forcing contributions to the ocean model performance, rather than suggesting these to be valid configurations for operational oceanography in themselves.

The summary results in Fig. 12 shows that SST is improved in pCPL_RAD (MD = 0.76 K, RMSD = 1.18 K) relative to FOR_GL, and has similar performance to FOR_HI during daytime in particular. This shows some benefit of using the regional scale source of heat budget information and global-scale wind forcing. The quality of SST results is lower for pCPL_RAD than when coupling both radiation and wind speed in CPL_AO however. This highlights that ensuring that the ocean state is in balance with the atmosphere is also important, requiring that the near-surface winds are consistent with the near-surface stability driven by air-sea temperature differences for example. Some evidence of the relationship between SST and near-surface atmosphere conditions within the coupled system used in this study was discussed by Lewis et al. (2018b; see their Figure 14). In accordance with the review of Small et al. (2008) for example, they described how an increase in SST through ocean-atmosphere coupling in the NWS can produce less stable near-surface conditions, which can increase near-surface wind speeds (and vice-versa). Meroni et al. (2018) more formally quantified the spatial correlations between mesoscale SST and wind speed variability at high resolution in the Gulf of Lion, which in turn was shown to impact on the distribution of heavy

rain bands. The use of an external source of wind forcing in the partially coupled pCPL_RAD experiment here 'breaks' any such near-surface stability-wind feedback, and seems to reduce the quality of SST results relative to the fully coupled simulations CPL_AO and CPL_AOW.

The quality of simulated SST is markedly reduced in pCPL_WIN (MD = 1.96 K, RMSD = 2.56 K).  This demonstrates the combined detrimental impact of applying a relatively coarse-scale description of the surface radiation budget originating from a global-scale atmosphere and highly variable and biased surface winds originating from the regional atmosphere simulation. In addition, the ocean and atmosphere are no longer in balance through use of the mixed coupling approach with incomplete representation of feedbacks. This result also confirms that the improvement in SST found in FOR_HI relative to FOR_GL is driven predominantly by the differences to the surface heat budget  between the two sources of atmospheric forcing.

## 4 Conclusions

This paper has demonstrated that simulation of ocean temperature for the NWS is sensitive to the atmospheric forcing at the surface. Better agreement of simulated SST with observations has been found for a near-coastal environment through use of information from a convective-scale  resolution regional atmosphere simulation rather than using data from a global-scale NWP forecast as applied in most current operational ocean forecast systems.

A key difference in the insolation in the global and regional-scale atmosphere models comes from the explicit representation of convective clouds and their impacts on radiation. In addition to the increased spatial variability from the regional-scale atmosphere simulations, a mean reduced $Q_{SW}$ of order 7% across the Celtic Sea region has been found compared to the global-scale forcing. In these simulations, which had a positive SST initial bias, this reduction contributed to improved SST prediction. The near-surface winds also differ between the global NWP and  regional-scale atmospheric simulations both in the mean and their variability. The regional atmosphere model winds compare less well to the limited number of observations over the ocean. It is therefore concluded that the impact of wind forcing is of second order to the treatment of insolation on the quality of SST results.

The SST bias in near-coastal areas is further reduced using two-way coupling between the ocean and atmosphere and reduced further by including feedbacks with surface waves. Lewis et al. (2018b) for example demonstrated this to be a general result, and is thought to result from the consistent simulation of the ocean and atmosphere and representation of feedbacks across the surface. SST results were improved relative to observations at a number of near-coastal sites during other times of the year (e.g. Figs. 3 and 4 of Lewis et al., 2018b), noting the impact of wave coupling to be more important during an autumn experiment period than found for the July period considered here. In general, while CPL_AOW results incorporating wave feedbacks were improved relative to CPL_AO, the main impact coupling in this study originates from inclusion of atmosphere-ocean interaction.

Although unavailable for the period considered here, the recent implementation of the AMM15 regional ocean configuration for operational forecasting across the NWS (Tonani et al., 2019, *this issue*) will provide a consistent ocean analysis for use in

future studies in the region. This will substantially reduce the initial condition errors discussed in this study, and further work to examine the response to changing forcing with no initial condition bias is encouraged.

Given the sensitivity of ocean predictions to the surface forcing and coupling demonstrated here it is clear that more routine observations of the components of the surface energy and momentum budgets over the ocean would be of considerable value.

5   In particular, co-location of complimentary measurements of the ocean and atmospheric boundary layers should better enable a more complete representation of surface feedbacks, in order to evaluate and improve prediction systems. Given these are challenging environments for making observations, making more use of the scarce sources of information currently available to the meteorological and oceanographic research communities should also be encouraged as a component of regional model development across both disciplines. The use of fully coupled prediction systems for research provides a framework in which

10   to focus efforts on evaluating the interactions across the ocean surface, and to identify gaps in the current observational capability above and below the surface.

**Acknowledgements**

This work originated from initial studies and technical developments conducted under the Copernicus Marine Environment Monitoring Service (CMEMS) evolution project on Ocean-Wave-Atmosphere Interactions in Regional Seas (OWAIRS).

15   CMEMS is implemented by Mercator Ocean in the framework of a delegation agreement with the European Union.

The Western Channel Observatory is funded by the UK Natural Environment Research Council through its National Capability Long-term Single Centre Science Programme, Climate Linked Atlantic Sector Science (grant number NE/R015953/1).

We acknowledge the constructive contributions of two anonymous reviewers whose comments have substantially improved this paper.

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

| Run ID | Model system[1] | Atm. coupled? | Wave coupled? | Information on meteorological forcing / coupling of ocean | | | |
|---|---|---|---|---|---|---|---|
| | | | | Source | Grid resolution | MetUM config. | Frequency |
| **FOR_GL** | UKO3g | No | No | Global-scale MetUM NWP forecast | Approx. 17 km | GA6.1, GL6.1 (Walters et al., 2017) | Radiation: 180 min Winds: 60 min |
| **FOR_HI** | UKO3h | No | No | Regional uncoupled MetUM | Variable resolution, up to 1.5 km | RA1 (Bush et al., 2018) | All: 60 min |
| **CPL_AO** | UKC3ao | Yes | No | Regional coupled MetUM | | | |
| **CPL_AOW** | UKC3aow | Yes | Yes | | | | |

Table 1: Summary of ocean simulation experiments using forced mode and coupled systems. ([1] The model system names refer to model configurations documented by Lewis et al. (2018b).

| | Full domain, 30 June – 30 July 2014 | | | Celtic Sea region, 20 – 30 July 2014 | |
|---|---|---|---|---|---|
| Experiment | MD (K) | RMSD (K) | | MD (K) | RMSD (K) |
| FOR_GL | 0.73 | 1.41 | | 1.22 | 1.56 |
| FOR_HI | 0.40 | 1.27 | | 0.63 | 1.17 |
| CPL_AO | 0.26 | 1.21 | | 0.36 | 0.99 |
| CPL_AOW | 0.20 | 1.24 | | 0.22 | 0.99 |

Table 2: Summary of mean difference of SST (Model – Observation) and root mean square difference (RMSD) comparing each simulation experiment with observations. Statistics computed using observations across the full AMM15 domain through July 2014 and those using only observations in the Celtic Sea region (Fig. 1(b)) during the last 10 days of July 2014 are listed.

| | Net radiation, standard deviation (W m$^{-2}$) | | | | 10 m wind speed, standard deviation (m s$^{-1}$) | | |
|---|---|---|---|---|---|---|---|
| Experiment | Mean | Max | Min | | Mean | Max | Min |
| FOR_GL | 55 | 190 | 10 | | 1.33 | 1.88 | 0.82 |
| FOR_HI | 78 | 277 | 16 | | 1.57 | 2.13 | 1.15 |
| CPL_AO | 81 | 268 | 17 | | 1.56 | 2.23 | 1.08 |
| CPL_AOW | 79 | 274 | 15 | | 1.54 | 2.14 | 1.11 |

Table 3: Summary of mean, maximum and minimum values of the spatial standard deviation of net radiation and 10 m wind speed computed across the Celtic Sea between 20 and 30 July 2014 for each experiment.

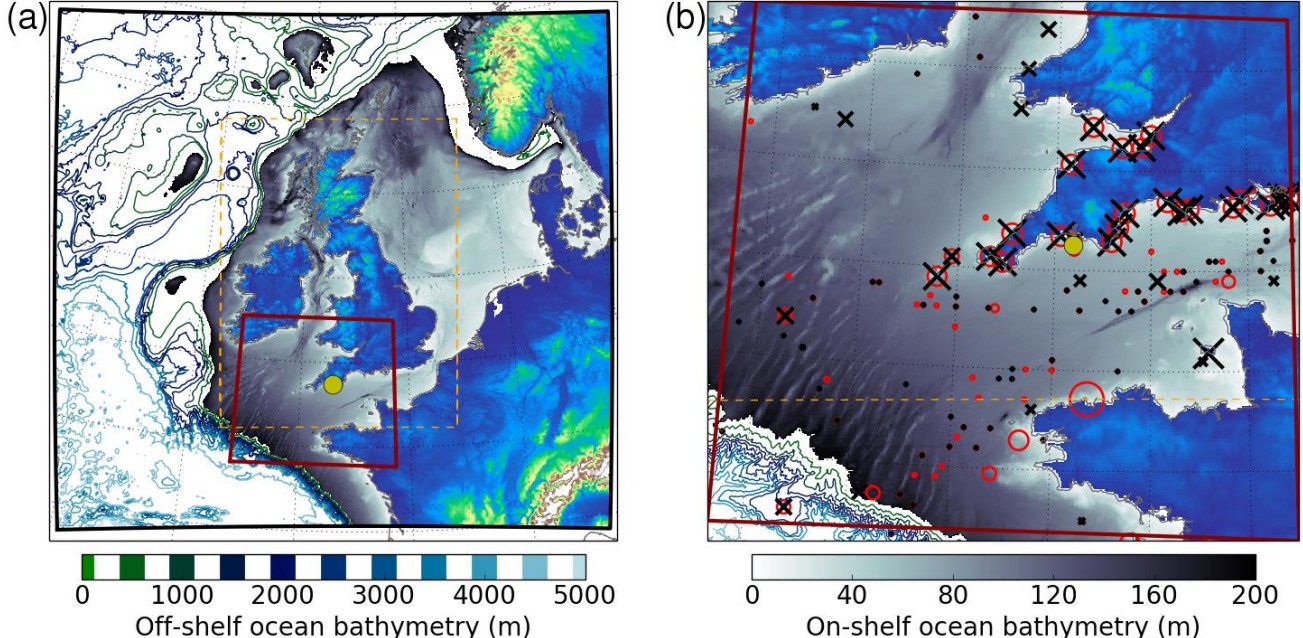

**Figure 1: (a) Regional ocean model bathymetry for the NWS system. The colour scale is valid for locations off the shallow shelf region. Also shown are the Celtic Sea study area (red box) and location of the L4 ocean buoy (yellow circle). The dashed orange area marks the inner region of the atmosphere model where grid cells are regularly spaced, becoming stretched outside this region. (b) Zoom in of ocean model bathymetry across the red box region (note on-shelf colour scale) Also shown are potential locations of in-situ observations of wind (black cross) and SST (red circle) available for evaluation between 20 and 30 July 2014. The size of symbols illustrates the volume of data at each location. The L4 ocean buoy is also shown as a yellow circle.**

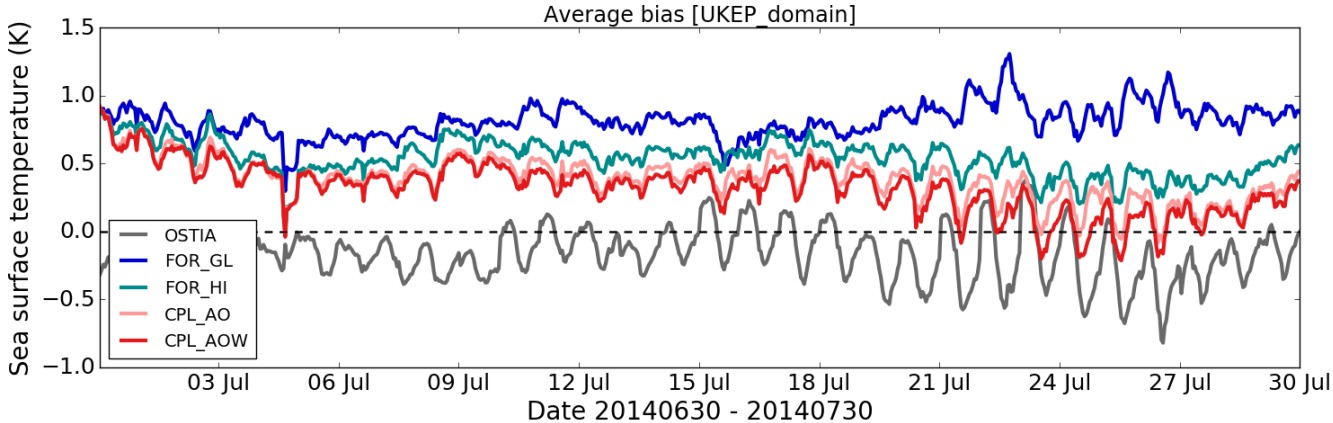

**Figure 2: Evolution of mean bias (Model – Observation difference) in SST for each experiment during July 2014 relative to all in-situ observations across the AMM15 model domain. Also shown is a comparison between daily OSTIA SST and in-situ observations.**

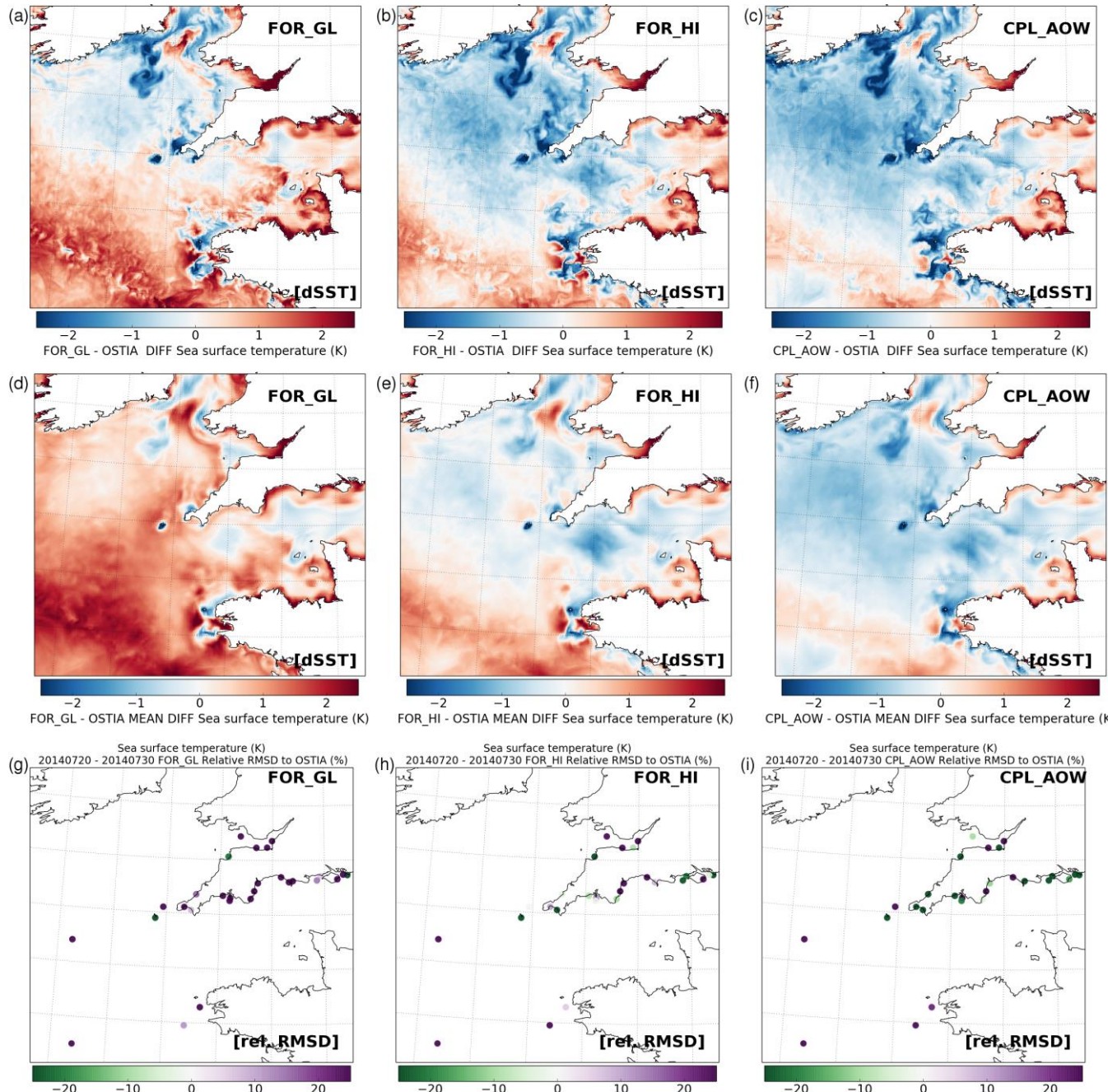

**Figure 3: (a-c)** Snapshot illustration of difference relative to OSTIA of the ocean model SST across Celtic Sea region valid at 1200 on 28 July 2014 from (a) FOR_GL configuration using global NWP forcing, (b) FOR_HI using 1.5 km resolution atmospheric forcing and (c) fully coupled CPL_AOW. **(d-e)** Mean difference of SST for each configuration relative to OSTIA over 10 day period between 20 and 30 July 2014. **(g-i)** Percentage change in RMSD comparing SST results with in-situ observations for (g) FOR_GL, (h) FOR_HI and (i) CPL_AOW results relative to RMSD between OSTIA and in-situ observations over this period.

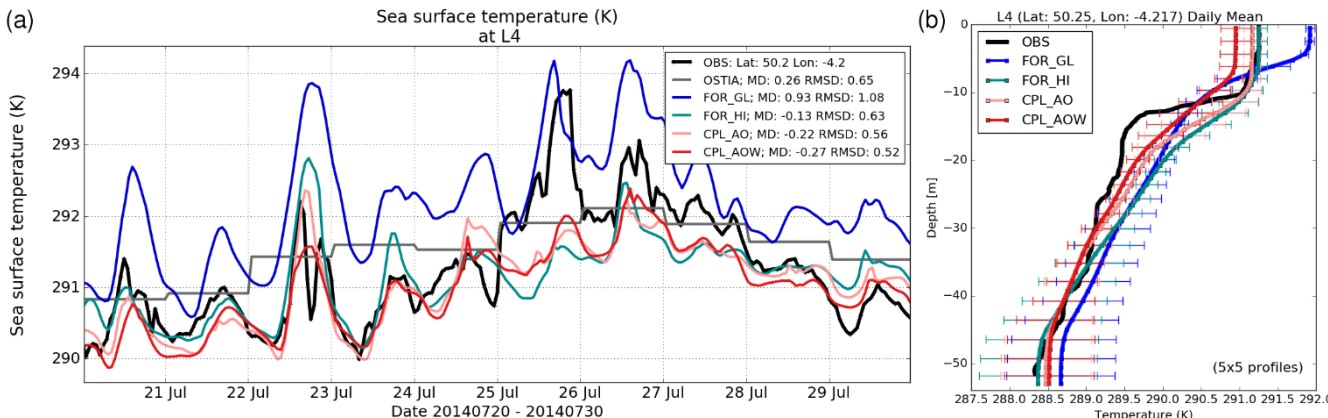

**Figure 4: (a) Time series of simulated and observed SST at the L4 ocean buoy (Fig. 1) between 20 and 30 July 2014. Model series are shown along with OSTIA as a mean from a 5 x 5 set of model grid cells nearest the observing site. (b) Vertical temperature profile observed by CDT at the L4 location on 28 July 2014 and daily mean profiles for each simulation experiment on that date. Error bars indicate 1 standard deviation around the spatial mean profile.**

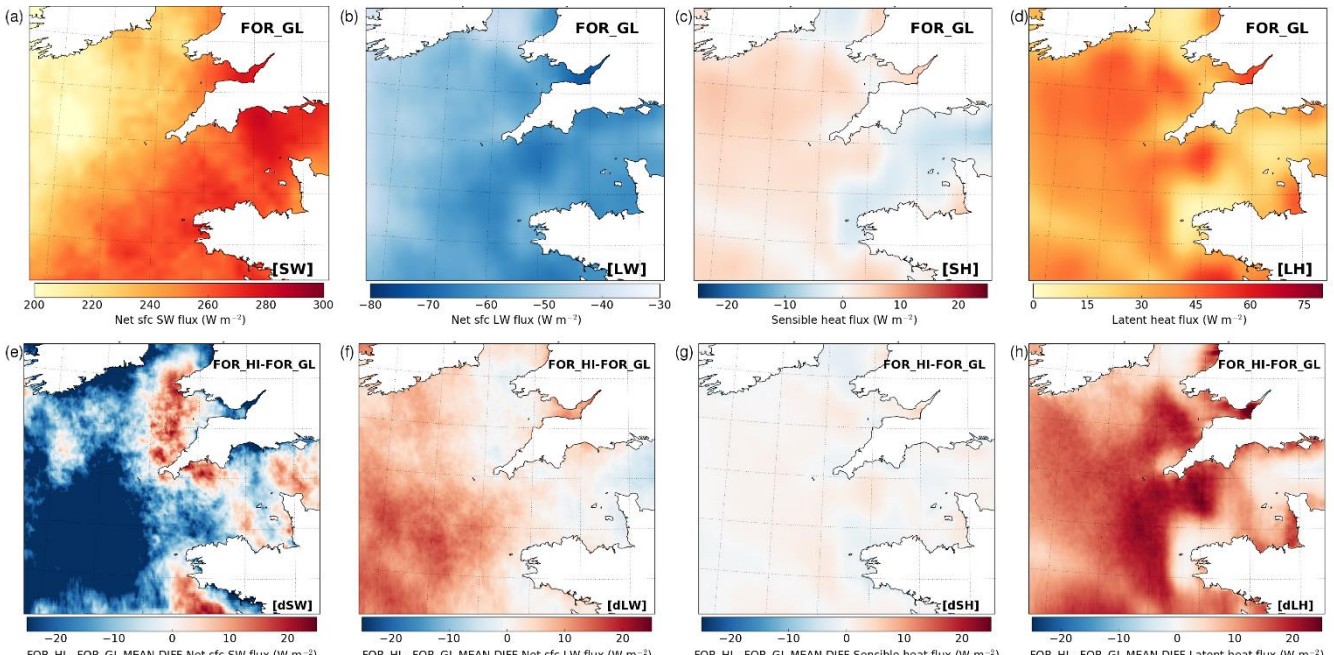

**Figure 5: Illustration of surface energy balance terms as 10-day mean from FOR_GL forcing between 20 and 30 July 2014, of (a) net surface downwelling shortwave flux, (b) net surface downwelling longwave flux, (c) sensible heat flux and (d) latent heat flux. Differences between FOR_HI and FOR_GL 10-day means for each variable are shown in (e)-(h).**

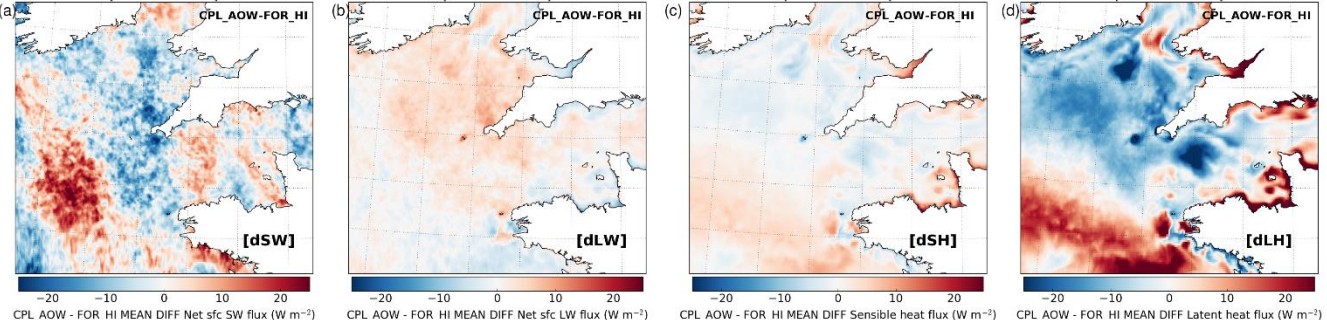

**Figure 6: The impact of model coupling across the Celtic Sea region shown as the difference between 10-day mean CPL_AOW and FOR_HI results across all times of day for (a) net surface downwelling shortwave flux, (b) net surface downwelling longwave flux, (c) sensible heat flux and (d) latent heat flux.**

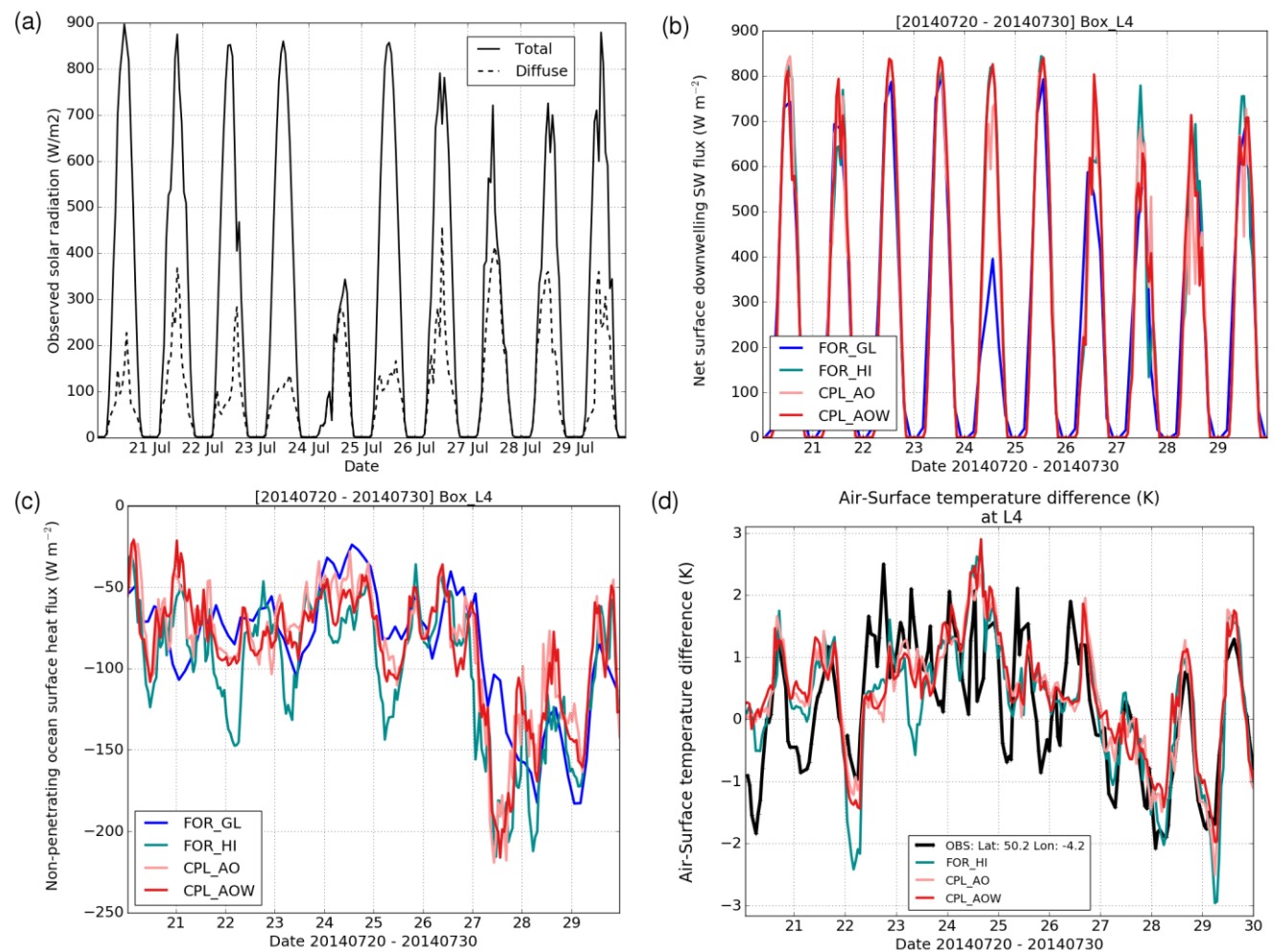

**Figure 7: (a)** Hourly mean observations of total and diffuse solar irradiance components at the L4 buoy between 20 and 30 July 2014. Time series of simulated **(b)** net surface downwelling shortwave flux, **(c)** non-penetrating ocean heat flux [Eq. 1] and () observations and simulations of near-surface temperature difference ($T_{air(1.5\,m)} - SST$).

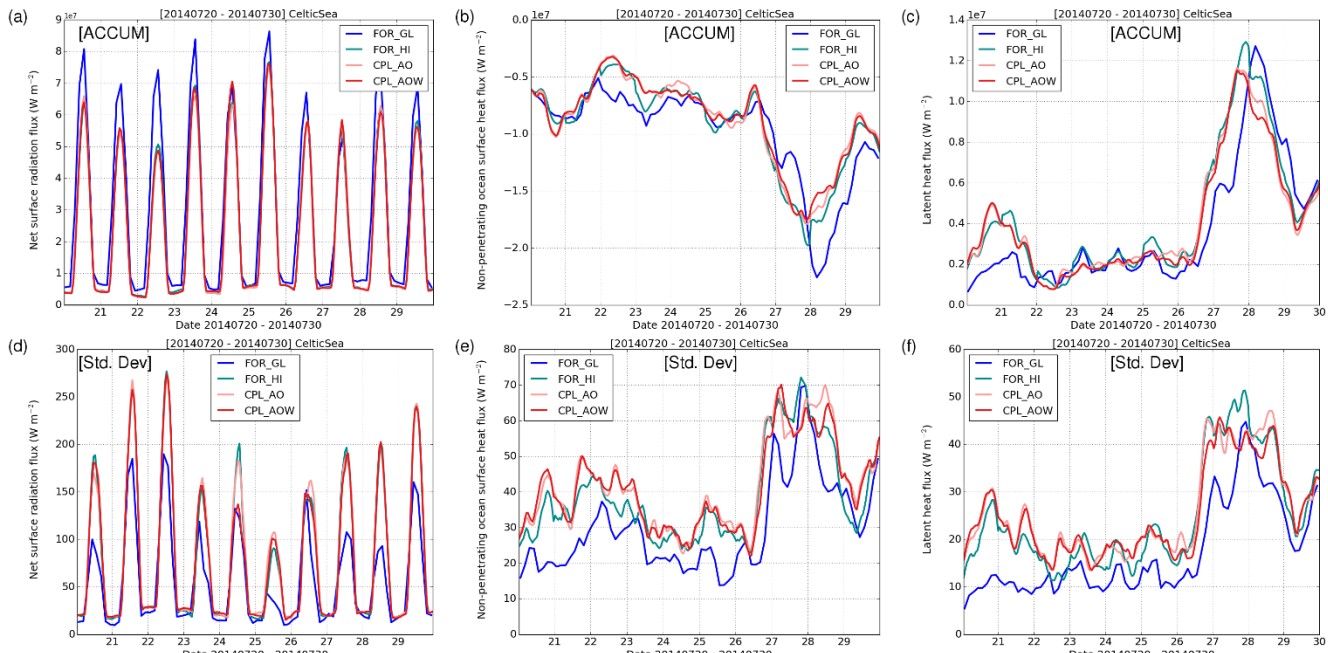

**Figure 8: Time series of simulated surface energy balance variables across sea areas in the Celtic Sea region (Fig. 1(b)), showing accumulations of (a) net surface radiation [net short-wave + net long-wave], (b) non-penetrating ocean heat flux [Eq. (1)], (c) latent heat flux, and time series of spatial standard deviations of (d) net surface radiation, (e) non-penetrating ocean heat flux and (f) latent heat flux across the region.**

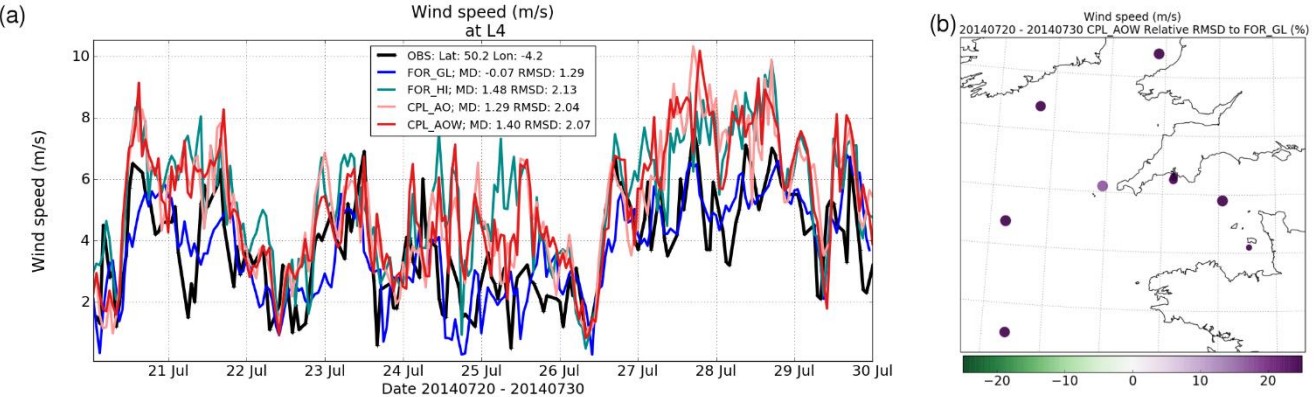

**Figure 9: Snapshot illustration of near-surface wind speed forcing across Celtic Sea region valid at 1200 on 28 July 2014 used for (a) FOR_GL configuration (global-scale NWP), (b) FOR_HI (1.5 km resolution atmosphere model) and (c) fully coupled CPL_AOW. Shaded circles show the distribution of instantaneous observed wind speed. (d) Mean near-surface wind speed forcing of FOR_GL over 10 day period between 20 and 30 July 2014, (e) 10-day mean of FOR_HI wind forcing, and (f) difference between 10-day mean of CPL_AOW with FOR_HI.**

**Figure 10: (a) Time series of simulated and observed near-surface wind speed at the L4 ocean buoy between 20 and 30 July 2014. Model series are shown as a mean from a 5 x 5 set of model grid cells nearest the observing site. (b) Percentage change in RMSD relative to in-situ observations for CPL_AOW wind speeds relative to FOR_GL forcing over this period.**

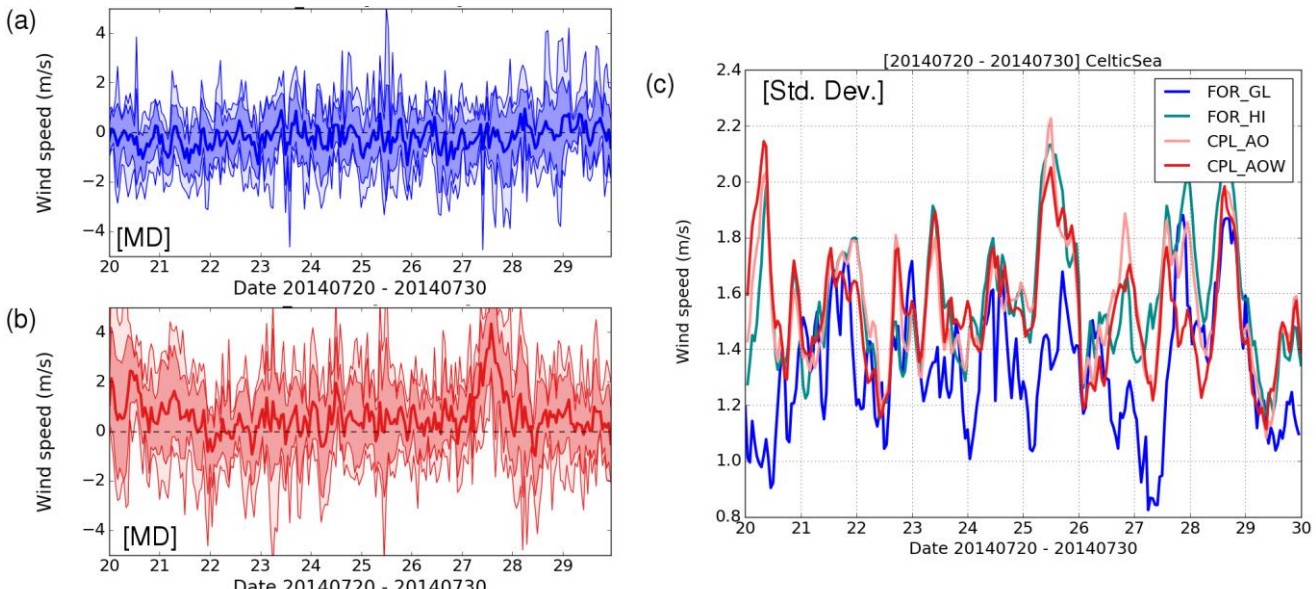

**Figure 11: Evolution of near-surface wind speed bias (model – observation) across Celtic Sea between 20 and 30 July 2014 for (a) FOR_GL forcing and (b) CPL_AOW simulation relative to in-situ observations. The mean bias across all sites is shown as a thick line, bounded by +/- 1 standard deviation (darker shading) and maximum/minimum differences (lighter shading). (c) Time series of the spatial standard deviation of simulated wind speed across Celtic Sea for each configuration.**

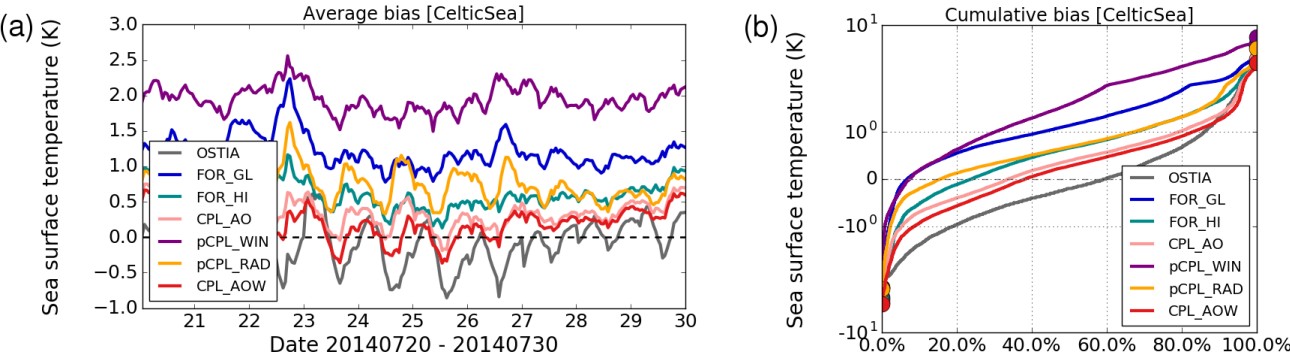

**Figure 12: (a) Evolution of bias (model – observations) in SST for all ocean forced, coupled and partially coupled experiments together with OSTIA data between 20 and 30 July 2014 relative to all in-situ observations across the Celtic Sea study area (red box in Fig. 1). (b) Cumulative SST bias distribution for each experiment.**