# Peer review of "Evaluating the impact of atmospheric forcing and air-sea coupling on near-coastal regional ocean prediction"

_Ocean Science, 2018_

## Referee Comment (RC1) · Anonymous Referee #1 · 4 Feb 2019

The study evaluates the impact of air-sea coupling in a North-West Europe shelf region and compares it with uncoupled simulations using high and low-resolution atmospheric forcing. It provides further evaluation of the coupled model system described in the GMD Discuss. paper by Lewis et al. with doi: 10.5194/gmd-2018-245. I got a slight impression, that the detailed evaluation of the improved system (UKC3) is separated into a second article to increase the number of publications (just using only the time period in summer and additional a slightly smaller area of interest). There is also an article about the evaluation of the wave coupling (doi: /10.5194/os-2018-148; did not read this one), which is introduced in the GMD discussion paper. This impression might be wrong, but it is also supported by the fact that the article reviewed here,

mainly uses the coupled experiment that also includes the wave model for evaluation of air-sea coupling. This evaluation is important and should be publish. However, I leave the decision whether to merge this article with the one submitted to GMD to the editor(s). In case this publication shall be a stand alone one, I suggest major revisions and would like to suggest a more detailed discussion of the changes in the physical processes (some are already there). Detailed comments are shown in the following.

Major comments:

I suggest to focus only on changes in model resolution and air-sea coupling here and mainly use the experiment CPL_AO and not CPL_AOW for comparison with uncoupled simulations (FOR_GL and FOR_HI) as CPL_AOW also includes the wave coupling. Although the CPL_AOW shows the best results, it would not be clear if differences arise from the air-sea coupling or the wave model coupling, especially as differences of wave model coupling are only shortly described in the section about near-surface wind speed. In addition, there is the other article submitted to Ocean Science about the wave coupling, which probably discuss this topic in detail.

The current article is structured in a way that it describes the different physical properties separately including all model experiments, which makes it difficult to get the connection of all changed processes. Therefore, I would like to suggest to change the structure of the results section of the article as follows and extent the physical analysis.

- The evaluation of the newly coupled system and the improvement compared to observations should be located in the GMD paper (is there also the Performance improvement between the UKC3 and UKC2 system shown?). Therefore, only a short overview should be given here (may also include the CPL_AOW simulation). - Then show how the uncoupled system changes with increasing resolution (FOR_GL vs. FOR_HI). Mention how the differences in the general physics influence the results (especially the atmospheric convection). Please describe here possible changes in the physical processes due to the increased resolution and how they influence the others (e.g. how

[Figure]

changes in radiation influence, SST, ocean currents, etc). What causes the changes? ... - Then compare the high resolution FOR_HI with the coupled system CPL_AO. How is the feedback of ocean to atmosphere changing wind speed and direction, and radiative fluxes. How are clouds influenced by changes in winds and how do they change the radiative fluxes. How is the ocean state changed e.g. how are ocean currents changed? Explain the origin of the larger scale patterns occurring in the differences in SST, wind and SW. When looking at these differences by eye, there might be similarities. If yes, what are their origins? Mechanism that shifts clouds leading to changes in radiation in the study area, explaining the differences in the mean fields and e.g. the biases in the SST? ...

Please also investigate the changes in physical processes in detail during the other seasons. Are there differences to the summer season?

Please make sure that in the conclusion and discussion, it is clear what the new finding of this study is compared to earlier studies and the GMD discussion paper.

There are some results "not shown", which should be. For example: - Page 8, Line 10: please show the examination of the cloud field - Page 9, line 16: please show the std - Page 9, line 25: please show the time-series of Qns

I would like to suggest to leave out section 3.4 (Partially coupled sensitivity experiments) as coupling only wind or radiation would lead to inconsistencies in the fluxes between atmosphere and ocean, which in my point of view would make interpretation difficult if not impossible.

During comparisons sometimes only snapshots of one particular time point is shown. Are these snapshots representative for the 10 day period?

Minor comments:

There are references used that are still in preparation, submitted or in review (Tonani, et al. 2018, Bush et al., 2018, Lewis et al., 2018c). I'm not sure if this is allowed in

Ocean Science. If they are published as discussion papers it might be possible, but what about the ones in preparation?

Page 2, line 17-20: Please make two sentences out of it as it it hard to read.

Page 2, line 21-23: how are the mesoscale ocean processes changed?

Page 2, line 28-29: Put here also other references (e.g. the examples used in the following)

Page 3, line 30: In this study: do you mean in the experiments FOR_GL and FOR_HI? Please clarify

Do atmosphere and ocean have the same model domain?

Page 4, line 19: please include a line break after "provided by Bush et al. (2018)."

Page 5, line 8-9: please provide a short description or a reference to the "double penalty" effects

Page 6, line 28-29, "potentially link" Please investigate if it is linked or not?

Page 7, line 8-11: Please also compare with model results in the morning to clarify if it is an artefact. Maybe also check it by using less neighborhood grid cells for averaging.

Page 7, line 12: Can you give a physical explanation for the improvement?

Page 8, line 20: cloud on 24 July → cloud cover on 24 July

Page 8, line 24: This could be related → Is this a hypothesis or is it related to

Page 9, line 14: with "patches" you mean the small scale spatial variations in the fields, right? If yes, do they origin from physical processes, (e.g. from sea surface roughness) or is it just the increased noise that occur in higher resolution models

Page 9, line 20-24: Please reduce the length of this sentence by separating it into at least two and eliminate "might be expected"

Page 9, line 32-33: For clarification, please reformulate the sentence

Page 12, line 4: further still by including → further by including

Please note if the forecasting system includes data assimilation. In case it does, mention the data that is assimilated in the method section.

Please mention in the method section, which parameters are used as atmospheric forcing for the uncoupled simulations and which parameters are exchanged between atmosphere and ocean and where the fluxes are computed.

In table 2 and table 3 and page 10, line 25: FIX_GL und FIX_HI written instead of FOR_GL and FOR_HI, please correct.

Please carefully check the reference list. For example, Valcke et al. (2015) is mentioned in the text but is missing in the reference list.

Figure 1a) please increase the lines in the colorbar to be able to identify the different colors

Figure 1b) what are the black and yellow dots? The Celtic Sea study are is partly located outside the UKV atmospheric grid. Does it impact the model results?

Figure 3) please use the same blue-red colorbar in a) and b) as used in c)

Figure 4a) and 4b) please increase the limits of the y-axis to include the max. difference in figure a). Also label the x-axis with the days (at the moment always 00.00)

In Labels of Figures 3, and 5-11 need to be larger to be readable in a printed copy.

---

## Referee Comment (RC2) · Anonymous Referee #2 · 19 Feb 2019

General comments:

The paper by Lewis et al. presents an evaluation of the impact of atmospheric forcing resolution (17km vs 1.5km) on a regional ocean forecast system (AMM15). The impact of high-resolution ocean-atmosphere(-waves) coupling is also investigated. The evaluation concerns the sea surface temperature with an important work of comparison to in-situ observations, then considering the heat budget and the wind modifications. The justification for only considering one month in summer is however missing.

Such kind of evaluation is quite often done in the regional climate modelling community

(e.g. Béranger et al., 2010; Akhtar et al. 2018a,b) and it could be relevant here to put forward the novelty of considering ocean forecasts with a very fine coupled system and the inherent difficulties. The added value of the wave coupling is also not so well highlighted.

My main remarks I briefly list here and detail more below concern:

- the key point in the ocean flux forcing which is the SST inconsistency between the one simulated in the ocean model and the one used at the surface boundary of the atmospheric model. It obviously controls the differences in several heat budget terms and very likely the differences in wind between CPL_xx and FOR(_HI) but it is never mentioned here.

- the robustness of the SST improvement (with the higher resolution forcing and coupling) that appears a little altered by the fact that it seems to be more a spin-up effect, with a reduction of the initial bias during the first days of simulation. In addition, the use of partially coupled sensitivity experiments seems very promising, but their results are too briefly discussed.

I'm finally interrogative about the large impact of the higher resolution which is always highlighted by the authors instead of the impact of the physics (qualified as a smaller impact). But for me this is connected, especially over sea, far offshore. I suggest to clarify or discuss more this point.

Consequently, I suggest some major revisions to improve the paper before accepting its publication.

Specific comments:

** Page 6, lines 2-3: "*This indicates improved SST prediction can be achieved for the*

*NWS when applying the high resolution.*"

In my opinion, this conclusion is too rapidly set. Figure 2 shows mostly the stronger cooling during the first days of simulation (till 5 July) in FOR_HI, CPL_AO and CPL_AOW, correcting more efficiently the initial warm bias. Considering the overestimation of the wind in the higher resolution forcing (coupling), this is a possible ocean response to the initial shock with a larger effect of the vertical mixing. How do the mixed layer's depth and thermal content evolve? If possible I suggest to test new initial conditions, more realistic, such as ocean analysis that are available in the CMEMS catalogue or at least a larger discussion about the relative importance of the forcing compared to the model initialisation.

Between 18 and 24 July, it seems there is a warming in FOR_GL whereas SST is stable in FOR_HI and CPL_xx. How is it explained?

** Page 8, lines 8 to 15: "*(. . .) The impact of coupling on (. . .) $Q_{LW}$ is dominated by random changes in the spatial distribution of convection. (. . .) There is also some evidence that the latent heat flux is increased in those near-coastal regions identified as being cooler in CPL_AOW than FOR_HI, where the coupled simulation SST was in closer agreement with observations in Fig. 3(c).*"

To well consider the differences in the heat budget terms between the coupled runs and FOR_HI, the comparison of the CPL_xx and OSTIA SST field(s) must be shown. I think it explains at first order most of the differences found in the long wave upward radiation, latent and sensible heat fluxes. The differences in the convective cloud location play also, but at a second order.

The last sentence is particularly confusing for me as it mixes information about LH, differences in SST simulated by CPL and FOR_HI and the validity of the CPL SST against observations. But what about the comparison between OSTIA and the in-situ observation in this region? Please revise.

Figure 6 (i-l): Please, adjust the scales to better show the differences. To be fair, it might be shown as relative differences (in %) instead.

Very likely, the differences in the wind field are also controlled by the differences in SST. See Chelton and Xie (2010) or for example Lebeaupin Brossier et al. 2015 (Fig. 8a), Meroni et al. 2018 (Fig. 6).

\*\* Page 9, lines 20-24: "*This provides some evidence that the differences between the representation of the surface energy budget in the global and regional-scale atmosphere simulations is driven mostly by the change in grid resolution and the change from parameterised to explicitly represented convection, rather than from differences between the underpinning MetUM radiation and cloud parameterisation choices, which might be expected to principally drive differences in the mean conditions rather than the spatial variability*"

Page 10, lines 14-16: "*The contrast between the spatial variability of wind speed between FOR_GL and FOR_HI further supports the assessment in Sect. 3.2 that the change in surface energy budget characteristics between the different sources of forcing were driven more by the change in atmosphere grid resolution than by changes to the underpinning model physics.*"

I can not really capture where the contribution of the high-resolution can be separated from the physical behaviour/parametrisations of MetUM between the FOR_GL and FOR_HI forcings. I mean, far from the coasts, there is no reason for these differences apart the MetUM physics?

In addition, connections between resolution and physics exist. Some physical parametrizations may depend on the grid resolution (and time step).

Please, clarify how you distinguish the relative importance of physics compared to the benefit of a finer grid mesh.

Other comments:

- p1, lines 14-15: "*Observations... data*". Please, revise the sentence as you do not only consider L4 observations...

- p1, lines 21-22: "*...by global-scale NWP (0.7 K in the model domain) is shown...*"

- p1, line 23: "*...reduced* (*by 0.2K*)."

- p2, lines 28-29: "*A number of studies... (Lewis et al., 2018a)*": revise citation.

- p3, line 7: "*...for one of those periods in July 2014.*" The motivation(s) to dedicate this study to this reduced period must be given here.

- p3, line 30: "*...describing the surface* heat and water *budget...*"

- p3, line 31 "*...NEMO using the 'flux formulation'...*": Where (and how) is computed the wind stress?

- p4, line 8 (and lines 17-18): "*the wave-dependent roughness Charnock parameter of 0.085 is used.*": Could you precise if it is $\alpha$ or $z_0$? If it is a constant, it is not wave dependent...?

- p4, line 17: "*...assumed zero and a constant value...*"

- p4, line 31: Valcke et al. (2015) is missing in the references list. Moreover, I think the citation for OASIS3-MCT is Craig et al., 2017.

- p4, line 31: "*...all information exchanged...*". A brief list of the exchanged fields could be useful.

[Figure]

- p5, lines 7-9: I am happy to see here this comment concerning the 'double penalty' effect that is indeed of primary importance when comparing high-resolution modelling results with observations.

- p5, end of section 2.2: I am a little surprise there are only GTS data considered. Some other kinds of data could be available on the CMEMS website, in particular profilers to examine the vertical stratification or satellite data that allows a 2D coverage at the surface. Was it a choice to exclude them? And if yes, why?

- p5, lines 26-27: "*Figure 2 demonstrates that all ocean simulations had the same initial conditions...*": This is not something that must be demonstrated. That must be said before in section 2.

- p6, line27-28: "*On several days (e.g. 20, 21, 23, 26 and 29 July) a tidally-forced heating signal of about 1 K is apparent.*" Well, it is not so apparent it is a tidally-induced heating or if it is a diurnal cycle.

- p6, lines 30-31: "*The SST variability of FOR_GL is in general stronger than observed...*" Where is it shown?

- p7, line 19: "*Surface* heat *budget*..." Please change also everywhere after: 'Energy' can be potential or kinetic. . . 'Heat' is more precise.

- p7, lines 29-30: "*...and from CPL_AO and CPL_AOW coupled systems...*": The flux fields for CPL_AO and CPL_AOW are not shown in Figure 6.

- p8, line 20: ". . . *increased cloud* cover *on 24 July...*"

- p8, lines 26-27: "*The warm surface temperature bias of FOR_GL at L4 appears to result despite rather than because of this difference however.*" Maybe mixing (*i.e.* cooling by entrainment) is also lower in FOR_GL?

- p9, line 8: "*...consistently higher during night time...*". Could you explain more this result?

- p9, lines 18-19: "*...both day and night...*" ?? "*...but typically of order 20-50% lower. . .*" Where the '50%' comes from?

- p9, line 27: Delete "*the accumulated* ".

- p10, line 17: "*The* atmospheric *forcing*..."

- p10, line 25: ". . . *than FIX_HI...*" FOR_HI?

- p11, lines 13-15: "*Some evidence of the link between SST and near-surface atmosphere conditions within the coupled system was discussed by Lewis et al. (2018b) in considering the relationship between near-surface stability and wind speed over the ocean.*" How this relates to the sentence before? More details or a summary of Lewis et al. (2018a)'s conclusions about the SST/stability/wind relationship would be useful.

- Tables 2/3: Replace FIX_xx by FOR_xx

- Please revise Figure 1: The colour scale for bathymetry in a is blank. What is the 'UKV' atmosphere grid? What are the small black and red points in b?

- Figure 2: If possible add the OSTIA SST error time-series.

- Figure 6: Please, adjust the colour bars in i, j, k, l.

- Figure 7: Add the colour legend for b (which simulation is the blue line?). Larger plots can also help to distinguish more the time-series in c.

- Figure 12: "*...between 20 and* 30 *July 2014...*"

Akhtar, N., Brauch, J. and Ahrens, B. (2018a): Climate modeling over the Mediterranean Sea: impact of resolution and ocean coupling. Clim. Dyn., 51: 933. https://doi.org/10.1007/s00382-017-3570-8.

Akhtar, N., Brauch, J. and Ahrens, B. (2018b): Erratum to: "Climate modeling over the Mediterranean Sea: impact of resolution and ocean coupling". Clim. Dyn., 51: 949. https://doi.org/10.1007/s00382-017-3678-x.

Béranger, K., Y. Drillet, M.-N. Houssais, P. Testor, R. Bourdallé-Badie, B. Alhammoud, A. Bozec, L. Mortier, P. Bouruet-Aubertot, and M. Crépon (2010): Impact of the spatial distribution of the atmospheric forcing on water mass formation in the Mediterranean Sea. J. Geophys. Res., 115, C12041, doi:10.1029/2009JC005648.

Chelton, D.B., and S.-P. Xie. (2010): Coupled ocean-atmosphere interaction at oceanic mesoscales. Oceanography, 23(4):52–69, doi:10.5670/oceanog.2010.05.

Craig, A., Valcke, S., and Coquart, L. (2017): Development and performance of a new version of the OASIS coupler, OASIS3-MCT_3.0. Geosci. Model Dev., 10, 3297–3308, https://doi.org/10.5194/gmd-10-3297-2017.

Lebeaupin Brossier, C., Bastin, S., Béranger, K. et al. (2015): Regional mesoscale air-sea coupling impacts and extreme meteorological events role on the Mediterranean Sea water budget. Clim. Dyn. 44: 1029. https://doi.org/10.1007/s00382-014-2252-z

Meroni, A. N., Parodi, A., and Pasquero, C. (2018): Role of SST patterns on surface wind modulation of a heavy midlatitude precipitation event. J. Geophys. Res. Atm., 123, 9081–9096. https://doi.org/10.1029/2018JD028276

———————————————

---

## Author Comment (AC1) · 12 Apr 2019

**Author response to RC1: "Evaluating the impact of atmospheric forcing resolution and air-sea coupling on near-coastal regional ocean prediction" by Huw W. Lewis et al.**

We thank RC1 for their constructive comments, which have led to improvements in the revised manuscript. Their contribution has also been acknowledged in the revised manuscript. A detailed response to the 'Major comments' and 'Minor comments' are provided further below.

RC1 also highlighted a particular concern, which we address directly, that:

*"….I got a slight impression, that the detailed evaluation of the improved system (UKC3) is separated into a second article to increase the number of publications (just using only the time period in summer and additional a slightly smaller area of interest….. There is also an article about the evaluation of the wave coupling (doi: /10.5194/os-2018-148; did not read this one), which is introduced in the GMD discussion paper…."*

The introduction text has been amended to be clearer about the distinction between the UKC3 description paper in GMD and this manuscript, but we also clarify the situation here.

The UKC3 system description paper has now been accepted for publication in GMD, and aimed to provide a high-level overview of system performance across 4 different times of year and evaluating results across atmosphere, ocean and wave components for the whole North West Shelf domain. The focus of that work is on the impact of coupling, and in particular on the effect of introducing new wave feedbacks within the ocean component in UKC3 (relative to UKC2 capability).

In contrast, this paper aims to take a much closer evaluation of the different atmospheric forcing on ocean results only, and to better define the impact of coupling we have conducted a series of new simulations not discussed at all in the GMD paper – referred to as FOR_HI, pCPL_WIN and pCPL_RAD here. The assessment of the FOR_HI results relative to FOR_GL is important to identify the impact of changing atmosphere forcing from global-scale to regional-scale, and therefore enables some measure of the additional impact of representing the feedbacks by then comparing coupled results with FOR_GL.

This paper focuses on only a small region to better highlight the changes in near-coastal regions, where we could make use of radiation measurements over sea at L4. A summer simulation period is selected as representing the period when atmosphere forcing and coupling changes had most impact. This also coincided with a period of good observational data coverage at L4, where we had access to both atmosphere and ocean observations. Of course, a more expansive discussion looking at a number of different times of year and locations within the domain would be desirable, but not feasible while also providing the kind of detailed evaluation advocated by RC1 as "*This evaluation is important and should be publish*".

In summary, the current manuscript is fundamentally different to the UKC3 system documentation paper, and no material initially intended for that paper has been "*separated*" into this paper as suggested. We therefore appreciate the editor continuing to consider the submitted paper for the CMEMS Special Issue, revised in light of reviewer comments.

The Ocean Science submission on the impact of wave coupling referenced by RC1, is also briefly cited in this paper, but concerns evaluation of wave impacts in the AMM15 system over a 2-year trial period (2017-2018) based on the operational ocean forecast configuration with/without data

assimilation, and using global-scale ECMWF forcing throughout. Beyond the common domain and use of the NEMO wave coupling configuration, there is very little practical overlap between the themes of the ocean-wave coupling paper and this manuscript.

***Author response to RC1 Major comments:***

1. *I suggest to focus only on changes in model resolution and air-sea coupling here and mainly use the experiment CPL_AO and not CPL_AOW for comparison with uncoupled simulations (FOR_GL and FOR_HI) as CPL_AOW also includes the wave coupling. Although the CPL_AOW shows the best results, it would not be clear if differences arise from the air-sea coupling or the wave model coupling, especially as differences of wave model coupling are only shortly described in the section about near-surface wind speed. In addition, there is the other article submitted to Ocean Science about the wave coupling, which probably discuss this topic in detail.*

We made a deliberate choice to show both CPL_AO and CPL_AOW results where possible, and consider the use of wave coupling to be important in general. The summary results (e.g. Fig. 2) show the differences between CPL_AO and CPL_AOW to be small at this time of year relative to the influence of including air-sea coupling, or the change in atmospheric forcing. Demonstrating this consistency is considered to be a useful result. The impact of wave coupling is discussed in more depth with regard to wind forcing, as the feedback between the wave model and atmosphere through the Charnock parameter has the potential to improve the wind forcing – here we argue that the SST-wind feedbacks are in fact more important, and that the coupling cannot 'correct' for the change in wind characteristics in the regional scale system relative to global.

As noted above, the 'other article submitted to Ocean Science about wave coupling' discusses a completely different experimental design over a two-year trial period with a focus on ocean results in the context of the operational CMEMS NWS system. Critically, there is no use of a regional scale atmosphere, or any ocean-atmosphere or wave-atmosphere feedbacks represented in the 'other article'.

2. *The current article is structured in a way that it describes the different physical properties separately including all model experiments, which makes it difficult to get the connection of all changed processes. Therefore, I would like to suggest to change the structure of the results section of the article as follows and extent the physical analysis.*
   a. *The evaluation of the newly coupled system and the improvement compared to observations should be located in the GMD paper (is there also the Performance improvement between the UKC3 and UKC2 system shown?). Therefore, only a short overview should be given here (may also include the CPL_AOW simulation).*
   b. *Then show how the uncoupled system changes with increasing resolution (FOR_GL vs. FOR_HI). Mention how the differences in the general physics influence the results (especially the atmospheric convection). Please describe here possible changes in the physical processes due to the increased resolution and how they influence the others (e.g. how C2 changes in radiation influence, SST, ocean currents, etc). What causes the changes? ...*

*c. Then compare the high resolution FOR_HI with the coupled system CPL_AO. How is the feedback of ocean to atmosphere changing wind speed and direction, and radiative fluxes. How are clouds influenced by changes in winds and how do they change the radiative fluxes. How is the ocean state changed e.g. how are ocean currents changed? Explain the origin of the larger scale patterns occurring in the differences in SST, wind and SW. When looking at these differences by eye, there might be similarities. If yes, what are their origins? Mechanism that shifts clouds leading to changes in radiation in the study area, explaining the differences in the mean fields and e.g. the biases in the SST? ...*

The current manuscript Results section is structured to

i)      Sect. 3.1: provide a brief overview of the SST results, setting the context of the experiments including FOR_HI and OSTIA results, which are not discussed at all in the GMD paper referenced.

ii)     Sect 3.2: discuss the evaluation of SST across the Celtic Sea sub-region and relative to the L4 observation point in particular, referenced later in the paper.

iii)    Sect 3.3: consider the different heat budget in all experiments.

iv)     Sect 3.4: consider the wind forcing in all experiments.
        Sect 3.5: explore partially coupled sensitivity experiments

We considered the suggestion of RC1 to effectively re-order the discussions of Sect 3.3 and 3.4 in particular to focus on both heat budget and wind forcing changes for a) FOR_HI vs FOR_GL, then b) CPL_AOW vs FOR_HI. On reflection we have kept the same overall structure however, noting it is helpful to compare all experiments relative to a particular observation or diagnostic (e.g. Fig. 7, 8, 10) within the same broad discussion. To first order, we also find that the heat budget and wind forcing of all regional scale simulations can be distinguished as a group from the FOR_GL forcing (e.g. Fig. 7, Fig. 10.). We have though revised these sections in light of the comment above, separating out more clearly the discussion of FOR_HI vs FOR_GL from CPL_AOW vs FOR_HI. In particular, note the revision of new Fig. 5 and new Fig. 6 to make this separation more explicit.

The discussion encouraged by RC1 in bullet c. above on the relationship between SST, wind and SW is better represented in the revised manuscript, including the encouragement of RC2 to consider the change in SST comparing OSTIA (used as a fixed daily boundary condition in FOR_GL and FOR_HI atmosphere model simulations) with CPL_AOW (e.g. Fig. 3f)) and its links to the changes in heat budget (e.g. Fig. 6d)) and wind (Fig. 9f) results.

*3. Please also investigate the changes in physical processes in detail during the other seasons. Are there differences to the summer season?*

As discussed above, we consider the extension of the current paper to results during other seasons to be out of scope, given the emphasis on a period when atmospheric forcing and coupling was considered to have largest effect. We better highlight this choice in the Introduction and Conclusion of the revised manuscript. Were we to add the suggested detailed analysis during other seasons, the manuscript would risk becoming overly long, and lose some of the detail requested by RC1 and RC2 in reviewing the current work.

4. *Please make sure that in the conclusion and discussion, it is clear what the new finding of this study is compared to earlier studies and the GMD discussion paper.*

The contrast to the GMD paper is explained more clearly in the revised Introduction. The first 3 paragraphs of the Conclusion refer to a summary of new findings of this study, and a characterisation of the sensitivity of ocean SST simulation over the NWS to choice of atmospheric forcing. The comparison to the GMD discussion paper is provided in the 4th paragraph.

5. *There are some results "not shown", which should be. For example: - Page 8, Line 10: please show the examination of the cloud field - Page 9, line 16: please show the std - Page 9, line 25: please show the time-series of Qns*

The cloud fields are not shown in interest of brevity, and as the fields are not readily available within the archived global-scale ocean model forcing fields (FOR_GL).

Other requested plots have now been added to the revised manuscript, within the updated Fig. 8 (std in d)-f)) and updated Fig. 7 (time series of Qns).

6. *I would like to suggest to leave out section 3.4 (Partially coupled sensitivity experiments) as coupling only wind or radiation would lead to inconsistencies in the fluxes between atmosphere and ocean, which in my point of view would make interpretation difficult if not impossible.*

RC1 is correct to highlight the inconsistencies in fluxes between atmosphere and ocean in the partially coupled sensitivity experiments. We highlight this point more clearly in the revised manuscript in light of this comment. However, given the encouragement of RC2 to expand rather than remove this section, we consider there still to be value in the results – indeed they help to illustrate the relative impact of the heat budget and wind forcing in isolation within the system, and enable us to conclude that coupling both wind and radiation leads to improved results although the evaluation of the regional-scale wind field is worse than the global scale atmosphere forcing.

7. *During comparisons sometimes only snapshots of one particular time point is shown. Are these snapshots representative for the 10 day period?*

Revised spatial plots in Fig. 3 (SST differences relative to OSTIA), Fig. 5 (heat budget terms and differences between FOR_HI and FOR_GL), Fig. 6 (differences in heat budget between CPL_AOW and FOR_HI) and Fig. 9d)-f) are all now consistently presented as 10-day means. This does mask some of the variability in fields such as the heat budget terms, but does provide a more representative illustration of results – indeed demonstrate that the snapshots presented in the original manuscript were generally representative.

***Author response to RC1 Minor comments***:

- *There are references used that are still in preparation, submitted or in review (Tonani, et al. 2018, Bush et al., 2018, Lewis et al., 2018c). I'm not sure if this is allowed in Ocean Science. If they are published as discussion papers it might be possible, but what about the ones in preparation?*

The references list has been amended with relevant doi to reflect the updated status of papers currently in review in Ocean Sciences. The only reference still listed as in preparation is Bush et al., (2018), which we anticipate to be submitted and have a citable doi very shortly and in time to be referenced were the current manuscript accepted for publication.

- *Page 2, line 17-20: Please make two sentences out of it as it it hard to read.*

This sentence has been shortened in the revised manuscript.

- *Page 2, line 21-23: how are the mesoscale ocean processes changed?*

An additional sentence has been added to more fully explain the results of Lebaupin Brossier et al. (2011).

- *Page 2, line 28-29: Put here also other references (e.g. the examples used in the following)*

In accordance with a related comment from RC2, the example citation has been updated and highlighted as offering a review of regional coupled studies in the revised manuscript.

- *Page 3, line 30: In this study: do you mean in the experiments FOR_GL and FOR_HI? Please clarify. Do atmosphere and ocean have the same model domain?*

Yes, and this is now clarified in the revised manuscript. The FOR_HI atmosphere and ocean model have the same model domain (as applied in coupled mode in CPL_AO and CPL_AOW simulations). This is also explicitly clarified in Section 2.1.

- *Page 4, line 19: please include a line break after "provided by Bush et al. (2018)."*

This is done in the revised manuscript.

- *Page 5, line 8-9: please provide a short description or a reference to the "double penalty" effects*

This has been added and a reference provided in the revised manuscript.

- *Page 6, line 28-29, "potentially link" Please investigate if it is linked or not?*

RC2 highlighted the difficulty in attributing the scale of SST variability to either the diurnal heating cycle or to tidal variations. In practice, the SST variability at any location will be some combination of these factors. We argue that the SST variability at L4 is mostly driven by the tides, but of course there is some influence of diurnal heating. It is not really possible to be any more definitive than 'potentially linked' in a paper of this scope, nor considered so important for the main conclusions drawn.

- *Page 7, line 8-11: Please also compare with model results in the morning to clarify if it is an artefact. Maybe also check it by using less neighbourhood grid cells for averaging.*

The vertical profile results presented in this study were available as daily mean diagnostics, so it is not easily practical to look at sub-daily patterns. The main aspect of interest from Fig. 4(b) is in contrasting the daily mean profiles from the four model experiments, using the CTD observations from L4 as a reference.

- *Page 7, line 12: Can you give a physical explanation for the improvement?*

We attribute the improvement to representing air-sea interactions within the coupled system, and the impact not only being apparent at the surface. Additional text has been added in the revised manuscript.

- *Page 8, line 20: cloud on 24 July → cloud cover on 24 July*

This has been amended, also in line with the related comment of RC2.

- *Page 8, line 24: This could be related → Is this a hypothesis or is it related to*

This is indeed a hypothesis. The text has been amended to be clearer.

- *Page 9, line 14: with "patches" you mean the small scale spatial variations in the fields, right? If yes, do they origin from physical processes, (e.g. from sea surface roughness) or is it just the increased noise that occur in higher resolution models*

Yes, we intended to say small scale spatial variations in the fields, although the 'patches' here refer to the variations where there is relatively reduced radiation. The text has been amended in the revised manuscript to clarify. We do not characterise source of the variations as "just the increased noise" but reflecting some combination of the explicit rather than parameterised convection, scale-dependent physics and different grid sizes in the higher resolution models. As discussed in response to RC2, attributing changes to resolution vs physics is a challenge.

- *Page 9, line 20-24: Please reduce the length of this sentence by separating it into at least two and eliminate "might be expected"*

The original sentence has been removed in light of the response to the comment above, and in line with the comments of RC2 in this regard.

- *Page 9, line 32-33: For clarification, please reformulate the sentence*

The sentence has been reviewed and reformulated as suggested.

- *Page 12, line 4: further still by including → further by including*

The sentence has been updated.

- *Please note if the forecasting system includes data assimilation. In case it does, mention the data that is assimilated in the method section.*

All regional-scale ocean and atmosphere simulations discussed in this paper are free-running without any data assimilation. This has been explicitly mentioned in the updated Section 2.

- *Please mention in the method section, which parameters are used as atmospheric forcing for the uncoupled simulations and which parameters are exchanged between atmosphere and ocean and where the fluxes are computed.*

This has been added in the updated Section 2, in line also with the comment of RC2.

- *In table 2 and table 3 and page 10, line 25: FIX_GL und FIX_HI written instead of FOR_GL and FOR_HI, please correct.*

Corrected.

- *Please carefully check the reference list. For example, Valcke et al. (2015) is mentioned in the text but is missing in the reference list.*

The reference list has been amended in the updated manuscript, and the missing Valcke et al. reference changed for a more recent reference to OASIS in light of RC2 comment.

- *Figure 1a) please increase the lines in the colorbar to be able to identify the different colors*

Corrected.

- *Figure 1b) what are the black and yellow dots? The Celtic Sea study are is partly located outside the UKV atmospheric grid. Does it impact the model results?*

The dots are indicative of the volume of data from each location during the period of interest, as now clarified in the updated figure caption. The yellow dot indicates the location of the L4 buoy. The caption has also been updated to clarify the 'outside the UKV atmospheric grid' – to highlight only that the inner region of the variable grid atmosphere domain has a regular spaced grid resolution, with stretching outside. We do not consider that this impacts the model results.

- *Figure 3) please use the same blue-red colorbar in a) and b) as used in c)*

In the original manuscript, Fig. 3c) presented model differences while a) and b) were model fields, so it was consistent to have different colorbars. In any case, Fig. 3 has been updated in the revised manuscript and its presentation and consistency of colorbars improved.

- *Figure 4a) and 4b) please increase the limits of the y-axis to include the max. difference in figure a). Also label the x-axis with the days (at the moment always 00.00)*

It was decided to merge some of the content between the original Fig. 3 and Fig. 4 in the revised manuscript given the suggested change in focus to consider spatial model fields relative to OSTIA also.

- *In Labels of Figures 3, and 5-11 need to be larger to be readable in a printed copy*

All Figures in the revised manuscript have been updated in light of this comment and in response to other reviewer comments.

---

## Author Comment (AC2) · 12 Apr 2019

**Author response to RC2: "Evaluating the impact of atmospheric forcing resolution and air-sea coupling on near-coastal regional ocean prediction" by Huw W. Lewis et al.**

We thank RC2 for their particularly constructive and detailed review comments and have amended the manuscript in response. Their contribution has also been acknowledged in the revised manuscript. In addition to correcting the list of 'Other comments' provided (see below), and a review of the full document in light of RC1 and RC2 comments in general, the following substantive changes have been addressed:

- Better justification for only considering one month in summer in this study,
- Improved linkage of relevance of this finer-scale ocean work with suggested references from the regional climate modelling community,
- More explicit reference to the positive but secondary added value of wave coupling,
- An updated presentation and discussion comparing SST simulations with OSTIA,
- Discussion of the initial condition bias, and opportunities for use of ocean analyses in future research,
- Better attribution of some heat budget differences to the different SST state in forcing and coupled atmosphere simulations,
- Expansion of the discussion on partially coupled results,
- More careful reference to resolution and physics changes between the global NWP and km-scale regional atmosphere forcing.

**Author Response to RC2 *General comments*:**

1. *The justification for only considering one month in summer is however missing.*

We agree, and have briefly provided a better explanation of the motivation in the last paragraph of the Introduction on p3. This change is also in line with a similar request from RC1 to more clearly articulate these choices. In brief, we selected to assess the July 2014 results and focus on a relatively small part of the model domain in order to provide a more detailed discussion of the impact of atmospheric forcing and coupling on near-coastal SST results, for a period identified by the overview discussion of Lewis et al. (2018b) as being most sensitive to coupling.

2. *Such kind of evaluation is quite often done in the regional climate modelling community (e.g. Béranger et al., 2010; Akhtar et al. 2018a,b) and it could be relevant here to put forward the novelty of considering ocean forecasts with a very fine coupled system and the inherent difficulties.*

The role of atmospheric forcing and coupling is indeed more routinely discussed in the context of typically coarser-scale regional climate modelling activities and it would be useful to contrast this with the present study. This has been addressed in an updated Introduction in the revised manuscript.

3. *The added value of the wave coupling is also not so well highlighted.*

We summarise in Sect. 3.1 that "there is some additional value evident from coupling information of the wave state to ocean and atmosphere components in CPL_AOW (MD = 0.20 K), although this is of secondary importance to the impact of either changing atmosphere resolution or ocean-atmosphere coupling", and in Section 3.3 that "the influence of wave coupling feedbacks is generally small at this time of year". The aim of presenting both CPL_AO and CPL_AOW results is both to demonstrate the performance of a fully coupled regional system, i.e. with wave coupling as an important component of the earth system at these scales, but also provide a more traceable comparison of the impact of coupling relative to the ocean-only results. This also addresses one of the comments of RC1. In light of the comment above, a new summary sentence has been added to the Conclusions to again highlight the relatively minor impact of wave coupling for this region and time of year.

4. *The key point in the ocean flux forcing which is the SST inconsistency between the one simulated in the ocean model and the one used at the surface boundary of the atmospheric model. It obviously controls the differences in several heat budget terms and very likely the differences in wind between CPL_xx and FOR(_HI) but it is never mentioned here.*

This is a fair challenge and an omission of the reviewed manuscript. The new comparison of SST against OSTIA (which provided the SST surface boundary of the atmospheric model) in the revised Figure 3 helps to highlight this point, and RC2 is correct to highlight the close spatial distributions of changes in SST and the change in sensible heat and latent heating in particular. This is now addressed in the revised manuscript in discussing the heat budget results of modified Fig. 6.

5. *The robustness of the SST improvement (with the higher resolution forcing and coupling) that appears a little altered by the fact that it seems to be more a spinup effect, with a reduction of the initial bias during the first days of simulation.*

We agree that longer-term simulations of the fully coupled simulation would be required to evaluate how robust the improvement is. However, we argue from Fig. 2 that the improvement becomes well established and is relatively constant by the second half of the month at least. This motivates us to discuss the 10 day period considered in most detail as being representative of a relatively steady state. Another interpretation of the comment on spinup, is that the simulations diverge relatively quickly in the first days of simulation, driven only by a change of atmospheric forcing or introduction of the atmosphere-ocean feedbacks.

6. *The use of partially coupled sensitivity experiments seems very promising, but their results are too briefly discussed.*

We appreciate this encouraging comment, and have provided some expansion of both the motivation for running the partially coupled sensitivity results and their assessment in the revised manuscript (Section 3.4). We balance this with addressing the concern of RC1 that they had less value given that the heat and wind terms are by definition not in equilibrium in these simulations. They aim to help better attribute the previous results described.

7. *I'm finally interrogative about the large impact of the higher resolution which is always highlighted by the authors instead of the impact of the physics (qualified as a smaller impact). But for me this is connected, especially over sea, far offshore. I suggest to clarify or discuss more this point.*

This is a valid concern, and a topic of discussion for the authors in the original assessment of the results in this study. The conflation of both resolution and physics changes between global NWP and what is characterised as the 'high resolution' forcing makes this a particular challenge. We have been encouraged by this comment to be more precise where possible in the revised manuscript as describing the global-scale and regional-scale forcing as being indicative of two readily available sources of atmosphere information, with the regional-scale also able to be applied with feedbacks. Changes have been made where relevant in the revised text. The paper title has also been updated in view of this comment.

The main reason for quoting the spatial grid resolution as dominating over physics changes originates from considering the larger spatial variability of forcing terms in FOR_HI than FOR_GL for example. We aim to be more careful in the revised manuscript that it is not clear we can attribute this directly to a resolution change alone.

***Author response to RC2 Specific comments:***

8. *Page 6, lines 2-3: "This indicates improved SST prediction can be achieved for the NWS when applying the high resolution." In my opinion, this conclusion is too rapidly set. Figure 2 shows mostly the stronger cooling during the first days of simulation (till 5 July) in FOR_HI, CPL_AO and CPL_AOW, correcting more efficiently the initial warm bias. Considering the overestimation of the wind in the higher resolution forcing (coupling), this is a possible ocean response to the initial shock with a larger effect of the vertical mixing. How do the mixed layer's depth and thermal content evolve?*

While the statement as written is correct (i.e. the SST Mean Difference for FOR_HI is lower than found for FOR_GL results), the tone of this line has been modified in the revised manuscript to be less definitive at that point, as we accept that it can be read as too definitive a conclusion.

The vertical profile in revised Fig. 4(b) shows the FOR_HI, CPL_AO and CPL_AOW simulations to have deeper mixed layers, consistent with RC2's comment, and with the persisting warm bias in FOR_GL. However, later discussion of the pCPL_RAD results show that by only applying the higher resolution (overestimated) winds does not diminish the warm bias in the same way, rather it increases over the first days of simulation and settles at order 2 K warmer than observations (Fig. 12). This further supports the value of the partially coupled simulations in drawing conclusions from the study (see response to comment 6 above).

9. *If possible I suggest to test new initial conditions, more realistic, such as such as ocean analysis that are available in the CMEMS catalogue or at least a larger discussion about the relative importance of the forcing compared to the model initialisation.*

We expand on this valid point in the revised Conclusions. While it is not practical to test new initial conditions in the present study (e.g. covering the period of interest), the relatively recent implementation of the 1.5 km resolution AMM15 ocean model configuration to provide CMEMS NWS MFC products (e.g. Tonani et al., 2018) does now offer a source of ocean analysis from the same system as used here, which should be valuable to support future research work.

10. *Between 18 and 24 July, it seems there is a warming in FOR_GL whereas SST is stable in FOR_HI and CPL_xx. How is it explained?*

The period highlighted by RC2 is apparent both for the domain-wide results in Fig. 2 and to some extent reflected in the location-specific comparison with observations at L4 in Fig. 4. While we do not provide detailed consideration of the evolution of FOR_GL results, the difference between FOR_GL and FOR_HI net shortwave radiation over period 20-30 July 2014 in the revised Fig. 5(e) highlights the relatively higher solar heating in FOR_GL described in the paper. Considering only the Celtic Sea region, the difference in Fig. 5(e) is focussed towards the south-west approaches, which coincides with the region of largest warm bias over the same period illustrated in the revised Fig. 3(d).

11. *Page 8, lines 8 to 15: "(. . .) The impact of coupling on (. . .) QLW is dominated by random changes in the spatial distribution of convection. (. . .) There is also some evidence that the latent heat flux is increased in those near-coastal regions identified as being cooler in CPL_AOW than FOR_HI, where the coupled simulation SST was in closer agreement with observations in Fig. 3(c)." To well consider the differences in the heat budget terms between the coupled runs and FOR_HI, the comparison of the CPL_xx and OSTIA SST field(s) must be shown. I think it explains at first order most of the differences found in the long wave upward radiation, latent and sensible heat fluxes. The differences in the convective cloud location play also, but at a second order. The last sentence is particularly confusing for me as it mixes information about LH, differences in SST simulated by CPL and FOR_HI and the validity of the CPL SST against observations. But what about the comparison between OSTIA and the in-situ observation in this region? Please revise.*

Comparisons between the FOR_GL, FOR_HI and CPL runs relative to OSTIA are now presented in a substantially revised Fig. 3. Time series comparisons relative to OSTIA are also now provided in Fig. 2 and Fig. 4 following this encouragement. The results discussion in Sect. 3 has been amended where relevant to describe these comparisons. We consider this provides a more coherent discussion than the original manuscript in line with the comment from RC2 above.

12. *Figure 6 (i-l): Please, adjust the scales to better show the differences. To be fair, it might be shown as relative differences (in %) instead. Very likely, the differences in the wind field are also controlled by the differences in SST. See Chelton and Xie (2010) or for example Lebeaupin Brossier et al. 2015 (Fig. 8a), Meroni et al. 2018 (Fig. 6).*

A version of Fig. 6 considering % differences was also prepared for the original manuscript, but changes were disproportionately dominated by regions where mean fluxes approached 0 $Wm^{-2}$. The impact of changing atmospheric forcing and coupling has now been separated (following RC1) across updated Fig. 5 and Fig. 6, where the comparison of heat budget terms are presented on a clearer

scale. We concur on the difference in wind field being controlled by differences in SST. See also RC2 comment and response on p11, line 13-15 below.

13. *Page 9, lines 20-24: "This provides some evidence that the differences…is driven mostly by the change in grid resolution and the change from parameterised to explicitly represented convection,…."*

    *Page 10, lines 14-16: "The contrast between the spatial variability of wind speed between FOR_GL and FOR_HI further supports the assessment in Sect. 3.2…"*

    *I cannot really capture where the contribution of the high-resolution can be separated from the physical behaviour/parametrisations of MetUM between the FOR_GL and FOR_HI forcings. I mean, far from the coasts, there is no reason for these differences apart the MetUM physics? In addition, connections between resolution and physics exist. Some physical parametrizations may depend on the grid resolution (and time step). Please, clarify how you distinguish the relative importance of physics compared to the benefit of a finer grid mesh.*

This reflect the RC2 Comment 7 discussed above, and is a valid query. Some of the key atmosphere physics differences are outlined in Sect. 2.1 As described in the response to Comment 7, the manuscript has been modified to take more care in describing the change of atmospheric forcing in terms of 'global-scale' and 'regional-scale', noting the link between grid resolution and physics choices. In particular, as noted in the paper, the main difference is in the treatment of convection explicitly at 1.5 km whereas it is parameterised in the global atmosphere model.

***Author response to RC2 Other comments:***

*• p1, lines 14-15: "Observations. . . data". Please, revise the sentence as you do not only consider L4 observations. . .*

Revised – we aim to highlight use of both the 'routine' operational observations along with use of L4 as having co-located observations of atmosphere and ocean.

*• p1, lines 21-22: "...by global-scale NWP (0.7 K in the model domain) is shown..."*

Corrected in the revised manuscript.

*• p1, line 23: "...reduced (by 0.2K)."*

Corrected in the revised manuscript.

*• p2, lines 28-29: "A number of studies. . . (Lewis et al., 2018a)": revise citation.*

It is not clear what is intended by this request. The intention of this citation was to really indicate Lewis et al. (2018a) as a source of further references. We have modified the citation to reference the more obvious review paper by Pullen et al. instead, mentioned elsewhere in the Introduction. We hope this might be what was intended by RC2 here.

*• p3, line 7: "...for one of those periods in July 2014." The motivation(s) to dedicate this study to this reduced period must be given here.*

We agree, and have briefly provided a better explanation of the motivation in the last paragraph of the Introduction on p3. This change is also in line with a similar request from RC1 to more clearly articulate these choices.

*• p3, line 30: "...describing the surface heat and water budget..."*

Corrected

*• p3, line 31 "...NEMO using the 'flux formulation'...": Where (and how) is computed the wind stress?*

In the configuration used in this study, key_shelf is used in NEMO, and the wind stress is computed within NEMO based on 10 m wind components rather than applying the atmosphere model computed stress directly. This is clarified in the revised manuscript and a reference provided.

*• p4, line 8 (and lines 17-18): "the wave-dependent roughness Charnock parameter of 0.085 is used.": Could you precise if it is α or z0? If it is a constant, it is not wave dependent. . .?*

This is $\alpha$. This sentence has been revised to clarify we mean a constant value used.

*• p4, line 17: "...assumed zero and a constant value..."*

Coorrected

*• p4, line 31: Valcke et al. (2015) is missing in the references list. Moreover, I think the citation for OASIS3-MCT is Craig et al., 2017.*

This correction has been applied in the revised manuscript.

*• p4, line 31: "...all information exchanged...". A brief list of the exchanged fields could be useful.*

This has been clarified with additional text at the end of Section 2.1

*• p5, lines 7-9: I am happy to see here this comment concerning the 'double penalty' effect that is indeed of primary importance when comparing high-resolution modelling results with observations.*

This is indeed an issue for evaluation of all such systems, and thank you for the supportive comment.

*• p5, end of section 2.2: I am a little surprise there are only GTS data considered. Some other kinds of data could be available on the CMEMS website, in particular profilers to examine the vertical stratification or satellite data that allows a 2D coverage at the surface. Was it a choice to exclude them? And if yes, why?*

We have focussed the analysis in this paper on the Celtic Sea region, and aim to make most use of the L4 buoy observations given the rare co-location of ocean and atmosphere observations, along with the radiation measurements. We also considered the co-located CTD observations from 28 July sufficient to provide some indication of the vertical profile in this region. The in-situ data on the CMEMS website (e.g. http://www.marineinsitu.eu/dashboard/) are in general consistent with those displayed in Fig. 1. We appreciate the encouragement to compare SST results with OSTIA, based on satellite data, which are now included in the revised manuscript (e.g. Fig. 2, Fig. 3).

*• p5, lines 26-27: "Figure 2 demonstrates that all ocean simulations had the same initial conditions.." This is not something that must be demonstrated. That must be said before in section 2.*

This sentence has been updated in the revised manuscript, although we consider it useful to remind readers of this from Fig. 2, particularly given that the later analysis focusses on the later period when

the 4 experiments have diverged. The initialisation is indeed referenced in Section 2.1 to indicate all simulations have the same initial condition.

• *p6, line27-28: "On several days (e.g. 20, 21, 23, 26 and 29 July) a tidally-forced heating signal of about 1 K is apparent." Well, it is not so apparent it is a tidally-induced heating or if it is a diurnal cycle.*

This sentence has been revised to be 'tidally-dominated', while we agree there will be some influence of diurnal cycle at this time of year. The temperature range observed at L4 is large – greater than 1K on some days in fact, and there is observed evidence of 'double peaks' on some days through the series. We also consider the phasing of the time of maximum temperature to be progressively delayed from around noon on 20 July to late evening on 25 July for example.

• *p6, lines 30-31: "The SST variability of FOR_GL is in general stronger than observed..." Where is it shown?*

This sentence referred to the temporal variability of simulation results at L4 shown in the new Figure 4. In addition to being biased warm, the FOR_GL results show larger diurnal range than other simulations and than observed. This line has been revised in the updated manuscript to clarify that we mean temporal rather than spatial variability here.

• *p7, line 19: "Surface heat budget..." Please change also everywhere after: 'Energy' can be potential or kinetic. . . 'Heat' is more precise.*

This has been revised everywhere mentioned through the manuscript.

• *p7, lines 29-30: "...and from CPL_AO and CPL_AOW coupled systems...": The flux fields for CPL_AO and CPL_AOW are not shown in Figure 6.*

The comparison of coupled results with FOR_HI have been separated a little from the FOR_GL vs FOR_HI comparisons, following the suggestion of RC1, as reflected in splitting out new Figure 5 from new Figure 6. The manuscript has been revised to reflect the updated Figures, and the required correction identified here has been removed as part of this.

• *p8, line 20: ". . . increased cloud cover on 24 July..."*

This has been updated in the revised manuscript.

• *p8, lines 26-27: "The warm surface temperature bias of FOR_GL at L4 appears to result despite rather than because of this difference however." Maybe mixing (i.e. cooling by entrainment) is also lower in FOR_GL?*

Rather than offer a detailed discussion here, we are simply noting that the SST results cannot be well explained by looking at the local energy balance terms within a relatively small area around the L4 location, as shown in Fig. 7(b). Rather, the results become a little clearer when assessing the atmospheric forcing over the broader Celtic Sea region (Fig. 8). This sentence has been revised to clarify this.

• *p9, line 8: "...consistently higher during night time...". Could you explain more this result?*

The result described in line 7 and line 8 – i.e. higher net radation from FOR_GL (contributing to higher SST) is resolved later on p9 from around line 20, where we relate the mean differences to lower spatial variability (lower standard deviations). This section has been revised further in the updated manuscript to attempt to clarify these discussions, noting earlier comments.

*• p9, lines 18-19: "...both day and night..." ?? "...but typically of order 20-50% lower. . ." Where the '50%' comes from?*

As requested by RC1, the time series of spatial standard deviation plots are now included in a revised Figure 8. This illustrates the substantially reduced standard deviation of radiation in FOR_GL relative to other configurations. The difference between daytime maxima through the period shown is considered to be of order 20-50% reduced.

*• p9, line 27: Delete "the accumulated ".*

Figure 8 has been revised, and now provides results as accumulated heat budget terms. This provides a clearer illustration of the differences between FOR_GL and FOR_HI forcing than mean values.

*• p10, line 17: "The atmospheric forcing..."*

This has been corrected. The original intention of "ocean forcing" was "The forcing of the ocean….", but this suggestion is clearer.

*• p10, line 25: ". . . than FIX_HI..." FOR_HI?*

Corrected in the revised manuscript.

*• p11, lines 13-15: "Some evidence of the link between SST and near-surface atmosphere conditions within the coupled system was discussed by Lewis et al. (2018b) in considering the relationship between near-surface stability and wind speed over the ocean." How this relates to the sentence before? More details or a summary of Lewis et al. (2018a)'s conclusions about the SST/stability/wind relationship would be useful.*

This section has been developed further in the revised manuscript, noting in particular RC2's comment 6 noting this section was too briefly discussed in the original. In summary, we argue that maintaining a feedback between SST, near-surface stability and near-surface winds is required.

*• Tables 2/3: Replace FIX_xx by FOR_xx*

Thank you, this has been corrected in the updated manuscript.

*• Please revise Figure 1: The colour scale for bathymetry in a is blank. What is the 'UKV' atmosphere grid? What are the small black and red points in b?*

The original Figure 1 colour scale in (a) was attempting to reference the contour lines off-shelf. These have now been made thicker in the revised manuscript. The caption text has been updated to clarify what was meant by the 'UKV grid', and the size of symbols referenced in the caption – the small points indicating points where there are a limited number of observations available over the selected period.

*• Figure 2: If possible add the OSTIA SST error time-series.*

Thank you for this suggestion. The comparison between daily OSTIA SST with in-situ observations is now included in the revised Figure 2. The OSTIA SST error has a strong diurnal signal given that it is a daily SST product, but comparisons with in-situ observations are hourly to be consistent with the model vs observation comparison. OSTIA data have also now been used as a reference in the revised Figure 3, and an OSTIA SST time series at the L4 location has been added to the revised Figure 4(a).

*• Figure 6: Please, adjust the colour bars in i, j, k, l.*

Figure 6(i-l) have now been pulled out into a new Figure 6 in the revised manuscript, focussing only on the impact of coupling (CPL_AOW-FOR_HI), with revised colour bars and clearer plots.

• *Figure 7: Add the colour legend for b (which simulation is the blue line?). Larger plots can also help to distinguish more the time-series in c.*

Corrected in revised Figure 7(b), and updated Figure 7 to have larger and clearer plots.

• *Figure 12: "...between 20 and 30 July 2014..."*

Corrected in revised manuscript.

---

## Referee Report (RR1)

The paper of Lewis et al. evaluates the impacts of atmospheric forcing and of ocean-atmosphere(-waves) coupling on high-resolution ocean forecasts in the Channel and the Celtic Sea coastal areas.

In my opinion, the revised version is clearly improved.
Also, the author responses to the reviewers have clarified a lot of points.

In more details:
- The heat budget terms and differences are now more clearly analysed, in particular considering and showing the OSTIA SST role and the differences with it.
- The relative impact of a better solved atmospheric physics related or not to a higher resolution is more discussed. Considering this point, the small change done in the title is particularly relevant.
- The justification for only considering one month in summer is now better expressed.

I have a final comment concerning the description of the exchanged fields in CPL_AO and I understand that the wind stress is not computed in the same compartment than in FOR_HI. This difference is not really detrimental here (especially in summer). Nevertheless, I suggest to consider this point for further comparisons between coupled and forced modes.

Finally, considering all this, I suggest to accept the paper for publication in Ocean Science.